# TRANSFORMER-BASED TRANSFORM CODING

**Yinhao Zhu**[*]    **Yang Yang**[*]    **Taco Cohen**
Qualcomm AI Research[†]
{yinhaoz, yyangy, tacos}@qti.qualcomm.com

## ABSTRACT

Neural data compression based on nonlinear transform coding has made great progress over the last few years, mainly due to improvements in prior models, quantization methods and nonlinear transforms. A general trend in many recent works pushing the limit of rate-distortion performance is to use ever more expensive prior models that can lead to prohibitively slow decoding. Instead, we focus on more expressive transforms that result in a better rate-distortion-computation trade-off. Specifically, we show that nonlinear transforms built on Swin-transformers can achieve better compression efficiency than transforms built on convolutional neural networks (ConvNets), while requiring fewer parameters and shorter decoding time. Paired with a compute-efficient Channel-wise Auto-Regressive Model prior, our SwinT-ChARM model outperforms VTM-12.1 by 3.68% in BD-rate on Kodak with comparable decoding speed. In P-frame video compression setting, we are able to outperform the popular ConvNet-based scale-space-flow model by 12.35% in BD-rate on UVG. We provide model scaling studies to verify the computational efficiency of the proposed solutions and conduct several analyses to reveal the source of coding gain of transformers over Conv-Nets, including better spatial decorrelation, flexible effective receptive field, and more localized response of latent pixels during progressive decoding.

## 1 INTRODUCTION

Transform coding (Goyal, 2001) is the dominant paradigm for compression of multi-media signals, and serves as the technical foundation for many successful coding standards such as JPEG, AAC, and HEVC/VVC. Codecs based on transform coding divide the task of lossy compression into three modularized components: transform, quantization, and entropy coding. All three components can be enhanced by deep neural networks: autoencoder networks are adopted as flexible nonlinear transforms, deep generative models are used as powerful learnable entropy models, and various differentiable quantization schemes are proposed to aid end-to-end training. Thanks to these advancements, we have seen rapid progress in the domain of image and video compression. Particularly, the hyper-prior line of work (Ballé et al., 2018; Minnen et al., 2018; Lee et al., 2019; Agustsson et al., 2020; Minnen & Singh, 2020) has led to steady progress of neural compression performance over the past two years, reaching or even surpassing state-of-the-art traditional codecs. For example, in image compression, BPG444 was surpassed by a neural codec in 2018 (Minnen et al., 2018), and (Cheng et al., 2020; Xie et al., 2021; Ma et al., 2021; Guo et al., 2021; Wu et al., 2020) have claimed on-par or better performance than VTM (a test model of the state-of-the-art non-learned VVC standard).

One general trend in the advancement of neural image compression schemes is to develop ever more expressive yet expensive prior models based on spatial context. However, the rate-distortion improvement from context based prior modeling often comes with a hefty price tag[1] in terms of decoding complexity. Notably, all existing works that claimed on-par or better performance than VTM (Cheng et al., 2020; Xie et al., 2021; Ma et al., 2021; Guo et al., 2021; Wu et al., 2020) rely on slow and expensive spatial context based prior models.

---

[*]Equal contribution.

[†]Qualcomm AI Research is an initiative of Qualcomm Technologies, Inc.

[1]In the extreme case when a latent-pixel-level spatial autoregressive prior is used, decoding of a single 512x768 image requires no less than 1536 interleaved executions of prior model inference and entropy decoding (assuming the latent is downsampled by a factor of 16x16).

The development of nonlinear transforms, on the other hand, are largely overlooked. This leads us to the following questions: can we achieve the same performance as that of expensive prior models by designing a more expressive transform together with simple prior models? And if so, how much more complexity in the transform is required?

Interestingly, we show that by leveraging and adapting the recent development of vision transformers, not only can we build neural codecs with simple prior models that can outperform ones built on expensive spatial auto-regressive priors, but do so with smaller transform complexity compared to its convolutional counterparts, attaining a strictly better rate-distortion-complexity trade-off. As can be seen in Figure 1, our proposed neural image codec SwinT-ChARM can outperform VTM-12.1 at comparable decoding time, which, to the best of our knowledge, is a first in the neural compression literature.

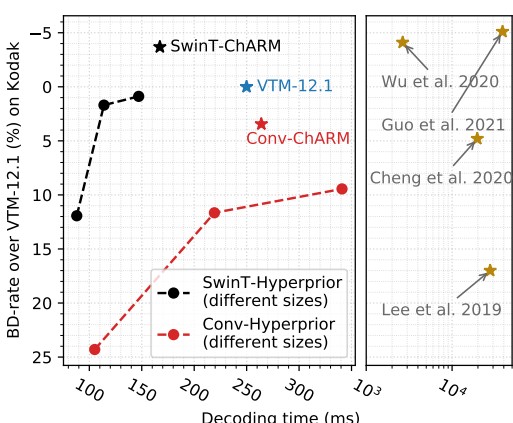

As main contributions, we 1) extend Swin-Transformer (Liu et al., 2021) to a decoder setting and build Swin-transformer based neural image codecs that attain better rate-distortion performance with lower complexity compared with existing solutions, 2) verify its effectiveness in video compression by enhancing scale-space-flow, a popular neural P-frame codec,

Figure 1: BD-rate (smaller is better) vs decoding time. Our Swin transformer based image compression models land in a favorable region of rate-distortion-computation trade-off that has never been achieved before. See Section 4.2 for more results and evaluation setup.

and 3) conduct extensive analysis and ablation study to explore differences between convolution and transformers, and investigate potential source of coding gain.

## 2 BACKGROUND & RELATED WORK

**Conv-Hyperprior** The seminal hyperprior architecture (Ballé et al., 2018; Minnen et al., 2018) is a two-level hierarchical variational autoencoder, consisting of a pair of encoder/decoder $g_a, g_s$, and a pair of hyper-encoder/hyper-decoder $h_a, h_s$. Given an input image $\mathbf{x}$, a pair of latent $\mathbf{y} = g_a(\mathbf{x})$ and hyper-latent $\mathbf{z} = h_a(\mathbf{y})$ is computed. The quantized hyper-latent $\hat{\mathbf{z}} = Q(\mathbf{z})$ is modeled and entropy-coded with a learned factorized prior. The latent $\mathbf{y}$ is modeled with a factorized Gaussian distribution $p(\mathbf{y}|\hat{\mathbf{z}}) = \mathcal{N}(\boldsymbol{\mu}, \text{diag}(\boldsymbol{\sigma}))$ whose parameter is given by the hyper-decoder $(\boldsymbol{\mu}, \boldsymbol{\sigma}) = h_s(\hat{\mathbf{z}})$. The quantized version of the latent $\hat{\mathbf{y}} = Q(\mathbf{y} - \boldsymbol{\mu}) + \boldsymbol{\mu}$ is then entropy coded and passed through decoder $g_s$ to derive reconstructed image $\hat{\mathbf{x}} = g_s(\hat{\mathbf{y}})$. The tranforms $g_a, g_s, h_a, h_s$ are all parameterized as ConvNets (for details, see Appendix A.1).

**Conv-ChARM** (Minnen & Singh, 2020) extends the baseline hyperprior architecture with a channel-wise auto-regressive model (ChARM)[2], in which latent $\mathbf{y}$ is split along channel dimension into $S$ groups (denoted as $\mathbf{y}_1, \ldots, \mathbf{y}_S$), and the Gaussian prior $p(\mathbf{y}_s|\hat{\mathbf{z}}, \hat{\mathbf{y}}_{<s})$ is made autoregressive across groups where the mean/scale of $\mathbf{y}_s$ depends on quantized latent in the previous groups $\hat{\mathbf{y}}_{<s}$. In practice, $S = 10$ provides a good balance of performance and complexity and is adopted here.

**Spatial AR models** Most of recent performance advancements of neural image compression is driven by the use of spatial auto-regressive/context models. Variants include causal global prediction (Guo et al., 2021), 3D context (Ma et al., 2021), block-level context (Wu et al., 2020), nonlocal context (Li et al., 2020; Qian et al., 2021). One common issue with these designs is that decoding cannot be parallelized along spatial dimensions, leading to impractical[3] decoding latency, especially for large resolution images.

---

[2]For details refer to Figure 11 and 12 in the Appendix.

[3]It is reported in (Wu et al., 2020)(Table I) that decoding time of spatial autoregressive models on a 512x768 image range from 2.6s to more than half a minute, depending on the specific designs. Also see Figure 1.

**ConvNet-based transforms** While the design space of prior models is extensively explored, nonlinear transforms, as an important component, have received less attention. A standard convolution encoder-decoder with GDN (Ballé et al., 2016; 2017) as activation is widely adopted in the literature. Later works introduce new transform designs, such as residual blocks with smaller kernels (Cheng et al., 2020), nonlocal (sigmoid gating) layers (Zhou et al., 2019; Chen et al., 2021), invertible neural networks (Xie et al., 2021), and PReLU as an efficient replacement of GDN (Egilmez et al., 2021).

**Vision transformers** Although many transform networks are proposed, they are still mainly based on ConvNets. Recently transformers (Vaswani et al., 2017) have been introduced to the vision domain and have shown performance competitive with ConvNets in many tasks, e.g. object detection (Carion et al., 2020), classification (Dosovitskiy et al., 2021), image enhancement (Chen et al., 2020), and semantic segmentation (Zheng et al., 2021). Inspired by their success, in this work we explore how vision transformers work as nonlinear transforms for image and video compression.

## 3 SWIN-TRANSFORMER BASED TRANSFORM CODING

Among the large number of vision transformer variants, we choose Swin Transformer (Liu et al., 2021) (hereafter referred to as SwinT) to build the nonlinear transforms, mainly because of 1) its linear complexity w.r.t. input resolution due to local window attention, and 2) its flexibility in handling varying input resolutions at test time, enabled by relative position bias and hierarchical architecture.

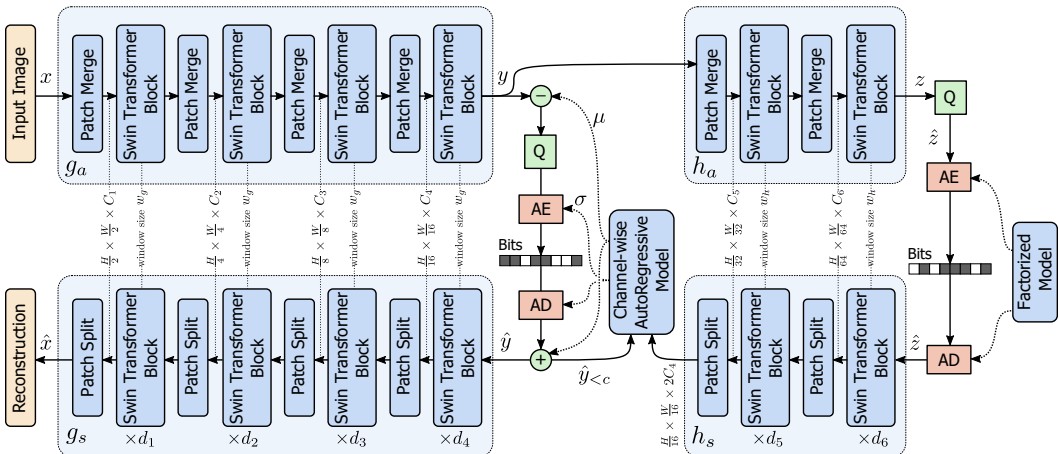

Figure 2: Network architecture of our proposed **SwinT-ChARM** model[4]. In the **SwinT-Hyperprior** model, the ChARM component is removed and instead $\mu$ and $\sigma$ are directly output by the hyperdecoder $h_s$ (For details see Figure 13 in Appendix A.1).

### 3.1 SWINT ENCODER AND DECODER

The original SwinT is proposed as a vision backbone, i.e. an encoder transform with downsampling. As shown in Figure 2, the SwinT encoder $g_a$ contains SwinT blocks interleaved with Patch Merge blocks. The Patch Merge block contains Space-to-Depth (for downsampling), LayerNorm, and Linear layers sequentially. SwinT block performs local self-attention within each non-overlapping window of the feature maps and preserves feature size. Consecutive SwinT blocks at the same feature size *shift* the window partitioning with respect to the previous block to promote information propagation across nearby windows in the previous block.

We adopt SwinT encoder as the encoder transform $g_a$ in our model, and extend it to SwinT decoder $g_s$ by reversing the order of blocks in $g_a$, and replacing the Patch Merge block with a Patch Split block, which contains Linear, LayerNorm, Depth-to-Space (for upsampling) layers in sequence. The architectures for hyper transforms $h_a, h_s$ are similar to $g_a, g_s$ with different configurations.

---

[4]The ChARM architecture (Minnen & Singh, 2020) is detailed in Figure 12 of Appendix A.1.

With these four SwinT transforms, we propose two image compression models, **SwinT-Hyperprior** and **SwinT-ChARM**, whose prior and hyper prior models are respectively the same as in Conv-Hyperprior and Conv-ChARM introduced in Section 2. The full model architectures are shown in Figure 2 and Figure 13.

## 3.2 EXTENSION TO P-FRAME COMPRESSION

To investigate the effectiveness of SwinT transforms for video compression, we study one popular P-frame compression model called Scale-Space Flow (SSF) (Agustsson et al., 2020). There are three instances of Conv-Hyperprior in SSF, which are respectively for compressing I-frames, scale-space flow and residual. We propose a SwinT variant, referred to as **SwinT-SSF**, which is obtained by replacing Conv transforms $g_a, g_s$ in the flow codec and residual codec of SSF with SwinT tranforms. To stabilize training of flow codec in SwinT-SSF, we need to remove all LayerNorm layers and reduce the window size (e.g. from 8 to 4). The baseline SSF model will be referred as Conv-SSF. Even though we build our solution on top of SSF, we believe this general extension can be applied to other ConvNet-based video compression models (Rippel et al., 2021; Hu et al., 2021) as well.

## 4 EXPERIMENTS AND ANALYSIS

### 4.1 EXPERIMENT SETUP

**Training** All image compression models are trained on the CLIC2020 training set. Conv-Hyperprior and SwinT-Hyperprior are trained with 2M batches. Conv-ChARM and SwinT-ChARM are trained with 3.5M and 3.1M steps. Each batch contains 8 random $256 \times 256$ crops from training images. Training loss $L = D + \beta R$ is a weighted combination of distortion $D$ and bitrate $R$, with $\beta$ being the Lagrangian multiplier steering rate-distortion trade-off. Distortion $D$ is MSE in RGB color space. To cover a wide range of rate and distortion, for each solution, we train 5 models with $\beta \in \{0.003, 0.001, 0.0003, 0.0001, 0.00003\}$. The detailed training schedule is in Appendix B.1.

For P-frame compression models, we follow the training setup of SSF. Both Conv-SSF and SwinT-SSF are trained on Vimeo-90k Dataset (Xue et al., 2019) for 1M steps with learning rate $10^{-4}$, batch size of 8, crop size of $256 \times 256$, followed by 50K steps of training with learning rate $10^{-5}$ and crop size $384 \times 256$. The models are trained with 8 $\beta$ values $2^\gamma \times 10^{-4} : \gamma \in \{0, 1, ..., 7\}$. We adopt one critical trick to stablize the training from (Jaegle et al., 2021; Meister et al., 2018), i.e. to forward each video sequence twice during one optimization step (mini-batch), once in the original frame order, once in the reversed frame order. Finally we add flow loss[5] only between 0 and 200K steps, which we found not critical for stable training but improves the RD.

**Evaluation** We evaluate image compression models on 4 datasets: Kodak (Kodak, 1999), CLIC2021 testset (CLIC, 2021), Tecnick testset (Asuni & Giachetti, 2014), and JPEG-AI testset (JPEG-AI, 2020). We use BPG and VTM-12.1 to code the images in YUV444 mode, and then calculate PSNR in RGB. For a fair comparison all images are cropped to multiples of 256 to avoid padding for neural codecs.

We evaluated P-frame models on UVG (Mercat et al., 2020) [6] and MCL-JCV (Wang et al., 2016), and compare them with the test model implementation of HEVC, referred to as HEVC (HM), and open source library implementation of HEVC, refered to as HEVC (x265). To align configuration, all video codecs are evaluated in low-delay-P model with a fixed GOP size of 12.

Besides rate-distortion curves, we also evaluate different models using BD-rate (Tan et al., 2016), which represents the average bitrate savings for the same reconstruction quality. For image codecs, BD-rate is computed for each image and then averaged across all images; for video codecs, BD-rate is computed for each video and then averaged across all videos. More details on testset preprocessing, and traditional codecs configurations can be found in Appendix B.2.

---

[5]We did not observe RD improvement when applying flow loss to Conv-SSF training.

[6]We use the original 7 UVG sequences that are commonly used in other works (Agustsson et al., 2020).

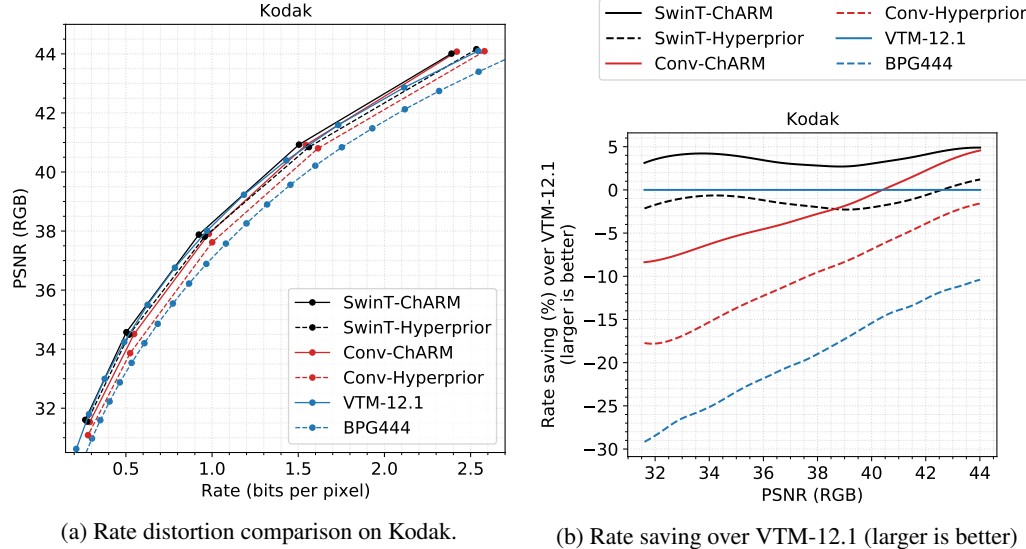

(a) Rate distortion comparison on Kodak.

(b) Rate saving over VTM-12.1 (larger is better)

Figure 3: Comparison of compression efficiency on Kodak[8]. Note that the encoding time of VTM-12.1 is much longer than all neural codecs, as shown in Table 4 and Table 7 in the Appendix.

## 4.2 RESULTS

**RD and BD-rate for image codecs** The RD curves for all compared image codecs evaluated on Kodak are shown in Figure 3a, and the relative rate reduction of each codec compared to VTM-12.1 at a range of PSNR levels is shown in Figure 3b [7].

As can be seen from Figure 3, SwinT transform consistently outperforms its convolutional counterpart; the RD-performance of SwinT-Hyperprior is on-par with Conv-ChARM, despite the simpler prior; SwinT-ChARM outperforms VTM-12.1 across a wide PSNR range. In the Appendix (Figure 28 and Figure 30), we further incorporate the results from existing literature known to us for a complete comparison. Particularly, our Conv-Hyperprior is much better than the results reported in (Minnen et al., 2018) (no context), and Conv-ChARM is on par with (Minnen & Singh, 2020).

| Image Codec | Kodak | CLIC2021 | Tecnick | JPEG-AI |
|---|---|---|---|---|
| BPG444 | 20.87% | 28.45% | 27.74% | 27.14% |
| Conv-Hyperprior | 11.65% | 12.23% | 12.49% | 20.98% |
| Conv-ChARM | 3.44% | 4.14% | 3.50% | 9.59% |
| SwinT-Hyperprior | 1.69% | 0.83% | -0.15% | 6.86% |
| SwinT-ChARM | **-3.68%** | **-5.46%** | **-7.10%** | **0.69%** |

Table 1: BD-rate of image codecs relative to VTM-12.1 (smaller is better).

In Table 1, we summarize the BD-rate of image codecs across all four dataset with VTM-12.1 as anchor. On average SwinT-ChARM is able to achieve 3.8% rate reduction compared to VTM-12.1. The relative gain from Conv-Hyperprior to SwinT-Hyperprior is on-average 12% and that from Conv-ChARM to SwinT-ChARM is on-average 9%. Further gain over VTM-12.1 can be obtained by test-time latent optimization (Campos et al., 2019) or full model instance adaptation (van Rozendaal et al., 2021), which are out of the scope of this work.

---

[7]The relative rate-saving curves in Figure 3b is generated by first interpolating the discrete RD points (averaged across the testset) with a cubic spline, and then compare bitrate of different models at fixed PSNR.

[8]RD plot for the other three datasets can be found in Appendix (Figure 14, Figure 15, and Figure 16)

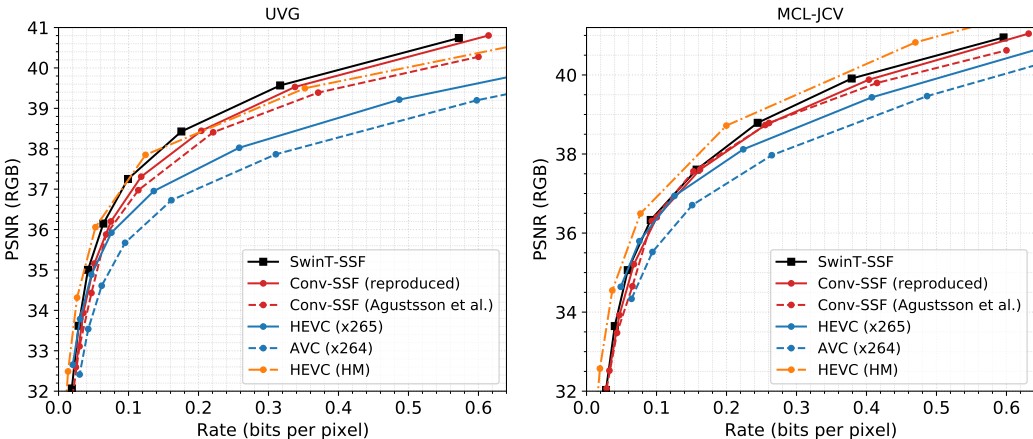

Figure 4: PSNR vs bitrate curves of P-frame codecs on UVG and MCL-JCV datasets.

**RD and BD-rate for video codecs**   For P-frame compression, we evaluated SwinT-SSF on UVG and MCL-JCV, with RD comparison shown in Figure 4. Again, SwinT transform leads to consistently better RD. Table 2 summarizes BD-rate with our reproduced Conv-SSF model as anchor. We can see that SwinT-SSF achieves an average of 11% rate saving over Conv-SSF. Additionally, we show that if SwinT transform is only applied to residual-autoencoder (labeled as SwinT-SSF-Res), it can only get about 4.6% gain, which indicates that both flow and residual compression benefit from SwinT as encoder and decoder transforms. Note that SwinT-SSF still lags behind HM, suggesting lots of room for improvement in neural video compression. For per-video breakdown of BD-rate, see Figure 18 and Figure 17 in the Appendix.

Table 2: BD-rate of video codecs, with Conv-SSF (reproduced) as anchor.

| Video Codec | UVG | MCL-JCV |
|---|---|---|
| HEVC (x265) | 25.97% | 25.83% |
| HEVC (HM) | **-15.80**% | **-24.96**% |
| SwinT-SSF | -12.35% | -10.03% |
| SwinT-SSF-Res | -5.08% | -4.16% |

Table 3: Decoding complexity. All models are trained with $\beta = 0.001$, evaluated on $768 \times 512$ images (average 0.7 bpp). Decoding time is broken down into inference time of hyper-decoder $h_s$ and decoder $g_s$, entropy decoding time of hyper-code $\hat{\mathbf{z}}$ and code $\hat{\mathbf{y}}$ (including inference time of the prior model if it is ChARM).

| Codec | Time (ms) | | | | | GMACs | Peak memory | Model params |
|---|---|---|---|---|---|---|---|---|
| | $\hat{\mathbf{z}}$ | $h_s$ | $\hat{\mathbf{y}}$ | $g_s$ | total | | | |
| Conv-Hyperprior | 5.5 | 4.0 | 38.2 | 168.9 | 219.3 | 350 | 0.50GB | 21.4M |
| Conv-ChARM | 5.3 | 4.1 | 82.9 | 168.5 | 264.0 | 362 | 0.53GB | 29.3M |
| SwinT-Hyperprior | 6.0 | 4.8 | 38.1 | 59.6 | 114.0 | 99 | 1.44GB | 24.7M |
| SwinT-ChARM | 5.9 | 4.8 | 90.7 | 60.1 | 167.3 | 111 | 1.47GB | 32.6M |

**Decoding complexity**   We evaluate the decoding complexity of 4 image codecs on 100 images of size $768 \times 512$ and show the metrics in Table 3, including decoding time, GMACs and GPU peak memory during decoding and total model parameters. The models run with PyTorch 1.9.0 on a workstation with one RTX 2080 Ti GPU. From the table, the inference time of SwinT decoder is less than that of Conv decoder. The entropy decoding time of ChARM prior is about twice than the factorized prior. The total decoding time of SwinT-based models is less than Conv-based models. In ablation study A5, we show a smaller SwinT-Hyperprior with 20.6M parameters has almost the same RD as the SwinT-Hyperprior profiled here. For details on encoding complexity, profiling setup, scaling to image resolution, please refer to Table 4 and Section D.3 in the Appendix.

**Scaling behavior** To see how the BD-rate varies with model size, we scale SwinT-Hyperprior and Conv-Hyperprior to be twice or half of the size of the base models (i.e. medium size)[9]. The result is shown in Figure 5. For both types of models, as we reduce the base model size, there is a sharp drop in performance, while doubling model size only leads to marginal gain. Noticeably, SwinT-Hyperprior-small is on-par with Conv-Hyperprior-medium even with half of the parameters, and SwinT transforms in general incur fewer MACs per parameter.

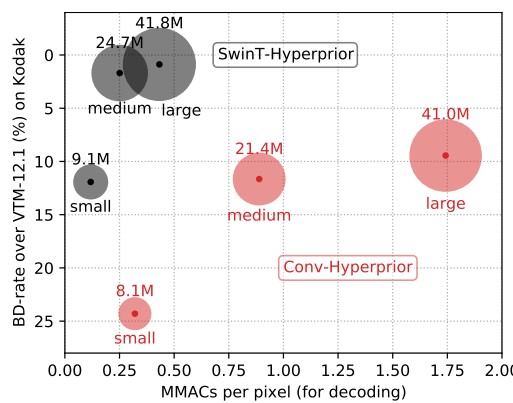

Figure 5: Model size scaling.

In Figure 1, we further consolidate the decoding latency and scaling behavior study into a single plot and show that SwinT-ChARM runs at comparable speed as VTM-12.1 while achieving better performance,[10] as opposed to state-of-the-art neural codecs with spatial autoregressive prior that decodes orders of magnitude slower.

### 4.3 ANALYSIS

**Latent correlation** One of the motivating principles of transform coding is that simple coding can be made more effective in the transform domain than in the original signal space (Goyal, 2001; Ballé et al., 2021). A desirable transform would decorrelate the source signal so that simple scalar quantization and factorized entropy model can be applied without constraining coding performance. In most mature neural compression solutions, uniform scalar quantization is adopted together with a learned factorized or conditionally factorized Gaussian prior distribution. It is critical, then, to effectively factorize and Gaussianize the source distribution so that coding overhead can be minimized.

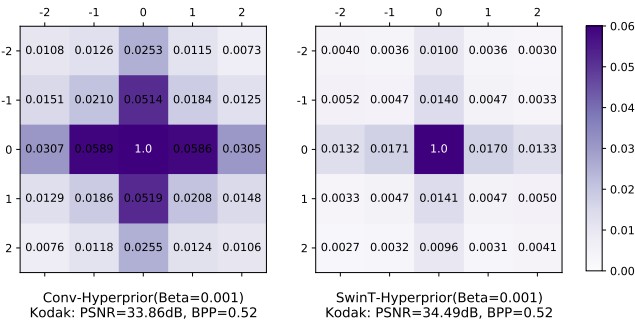

Figure 6: Spatial correlation[11] of $(\mathbf{y} - \boldsymbol{\mu})/\boldsymbol{\sigma}$ with models trained at $\beta = 0.001$. SwinT-Hyperprior (right) achieves uniformly smaller correlation than Conv-Hyperprior (left).

Specifically, in hyperprior based models (Ballé et al., 2018), $\bar{\mathbf{y}} \triangleq (\mathbf{y} - \boldsymbol{\mu})/\boldsymbol{\sigma}$ is modeled as a standard spherical normal vector. The effectiveness of the analysis transform $g_a$ can then be evaluated by measuring how much correlation there is among different elements in $\bar{\mathbf{y}}$. We are particularly interested in measuring the correlation between nearby spatial positions, which are heavily correlated in the source domain for natural images. In Figure 6, we visualize the normalized spatial correlation of $\bar{\mathbf{y}}$ averaged over all latent channels, and compare Conv-Hyperprior with SwinT-Hyperprior at $\beta = 0.001$. It can be observed that while both lead to small cross-correlations, Swin-Transformer does a much better job with uniformly smaller correlation values, and the observation is consistent

---

[9]detailed model configurations are provided in Appendix A.3.

[10]Note that it is difficult to fairly compare the decoding time of VTM and neural codecs since they run on different hardware. For more discussion please refer to Appendix D.3.

[11]The value with index $(i, j)$ corresponds to the normalized cross-correlation of latents at spatial location $(w, h)$ and $(w + i, h + j)$, averaged across all latent elements of all images on Kodak.

with other $\beta$ values, which are provided in Figure 20 in the Appendix. This suggests that transformer based transforms incur less redundancy across different spatial latent locations compared with convolutional ones, leading to an overall better rate-distortion trade-off. The larger spatial correlation (and thus redundancy) in Conv-Hyperprior also explains why a compute-heavy spatial auto-regressive model is often needed to improve RD with convolutional based transforms (Minnen et al., 2018; Lee et al., 2019; Ma et al., 2021; Guo et al., 2021; Wu et al., 2020). Figure 6 also reveals that most of the correlation of a latent comes from the four elements surrounding it. This suggests that a checkerboard-based conditional prior model (He et al., 2021) may yield further coding gain.

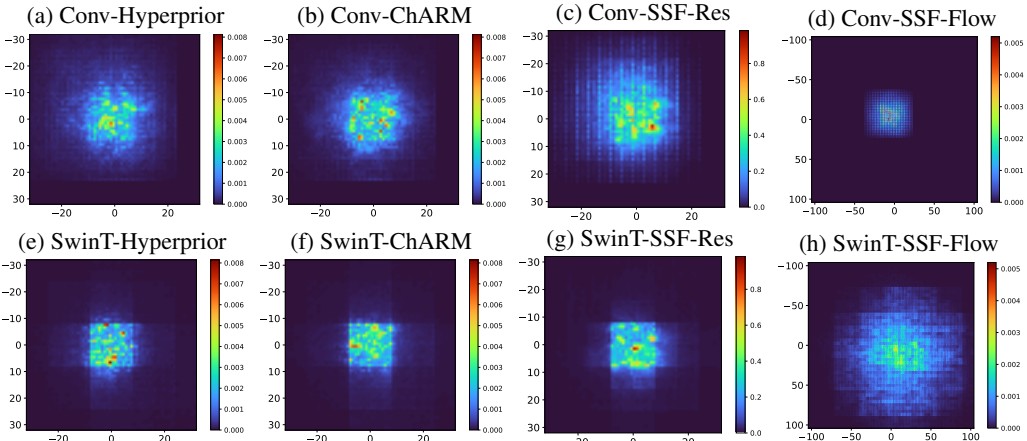

Figure 7: Comparison of effective receptive field (ERF) of the encoders $g_a$, which is visualized as absolution gradients of the center pixel in the latent (i.e. $d\mathbf{y}/d\mathbf{x}$) with respect to the input image. The plot shows the close up of the gradient maps averaged over all channels in each input of test images/videos. Check Figure 21 for the ERF of the composed encoding transform $h_a \circ g_a$.

**Effective receptive field** Intra prediction in HEVC or AV1 only rely on left and top boarders of the current coding block (Sullivan et al., 2012; Chen et al., 2018), except for intra block copy for screen content (Xu et al., 2016). We would like to see how large the effective receptive field (ERF) (Luo et al., 2017) of SwinT encoders compared to Conv encoders. The theoretical receptive field of the encoders ($g_a, h_a \circ g_a$) in SwinT-based codecs is much larger than that of Conv-based codecs. However comparing Figure 7a with 7e and Figure 7b with 7f, the ERF of SwinT encoders after training is even smaller than Conv encoders. When we examine the ERF of the released Swin transformers for classification, detection and segmentation tasks, they are all spanning the whole input image. This contrast suggests that (natural) image compression with rate-distortion objective is a *local* task, even with transformer-based nonlinear transforms. We further look into P-frame compression models, particularly the ERF of two types of transforms in flow codec and residual codec, as shown in Figure 7d & 7h, and Figure 7c & 7g. Clearly for flow codec, SwinT transform has much larger ERF than the convolution counterpart. For residual codec, the ERF of SwinT transforms is similar to image (I-frame) compression case. This shows of flexibility of SwinT encoders to attend to longer or shorter range depending on the tasks. To get a better picture of the behavior of attention layers in SwinT transforms, we also show the attention distance in each layer in Figure 22.

**Progressive decoding** The ERF in the previous section shows the behavior of the encoder transforms, here we further investigate the decoder transforms through the lens of progressive decoding (Rippel et al., 2014; Minnen & Singh, 2020; Lu et al., 2021). Initialized with the prior mean, the input to the decoder is progressively updated with the dequantized latent $\hat{\mathbf{y}}$ in terms of coding units, leading to gradually improved reconstruction quality. For the latent with shape $(C, H, W)$, we consider three types of coding units, i.e. per channel $(1, H, W)$, per pixel $(C, 1, 1)$, per element $(1, 1, 1)$. The coding units are ordered by the sum of prior std of all elements within each unit. The RD curves of progressive decoding for SwinT-Hyperprior and Conv-Hyperprior are shown in Figure 8a, which closely follow each other when ordered by channel or element, but significantly apart when ordered by pixel (spatial dim). Particularly, we show an extreme case when the half pixels in the latent (masked by checkerboard pattern) updated with dequantized values, corresponding to the two scatter points in Figure 8a. One visual example (CLIC2021 test) is shown in Figure 8b under

this setup, where we can clearly see SwinT decoder achieves better reconstruction quality than the Conv decoder, mainly in terms of more localized response to a single latent pixel. This is potentially useful for region-of-interest decompression. More visual examples are shown in Figure 26.

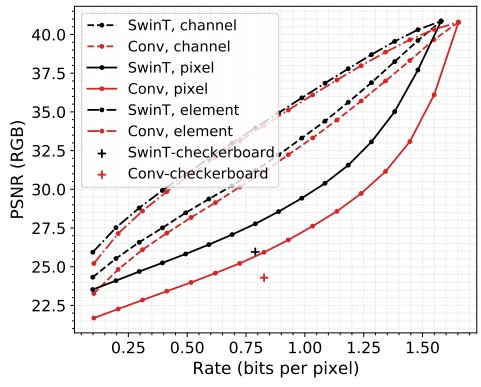

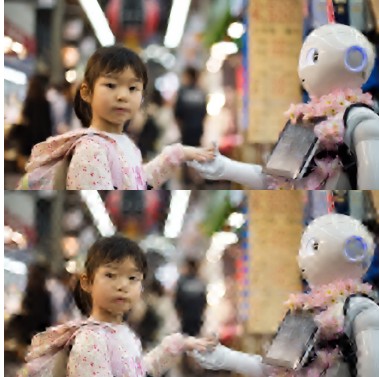

(a) Progressive decoding.

(b) Recon with checkerboard masked latent.

Figure 8: (Left) Progressive decoding according to the order of the sum of prior std of all elements within each latent coding unit, which can be one latent channel, pixel or element. SwinT-Hyperprior archives about 2dB better reconstruction at the same bitrate than Conv-Hyperprior for per-pixel progressive decoding. (Right) The reconstructions with SwinT-Hyperprior (top) and Conv-Hyperprior (bottom) for the latent masked with checkerboard pattern corresponding to the two plus markers on the left subfigure. The SwinT decoder has more localized reconstruction to a single latent pixel.

## 4.4 ABLATION STUDY

**Relative position bias** There are two sources of positional information in SwinT transforms, namely the Space-to-Depth modules and the additive relative position bias (RPB). Even when the RPB is removed, SwinT-Hyperprior still outperforms Conv-Hyperprior across all bitrates, which indicates image compression may not require accurate relative position.

**Shifted window** The original motivation of shifted window design is to promotes the inter-layer feature propagation across non-overlapping windows. Image compression performance drops slightly when there is no shifted window at all. This further suggests image compression requires local information.

The details of ablations A3-A5 in Figure 9 can be found in Section F of the appendix.

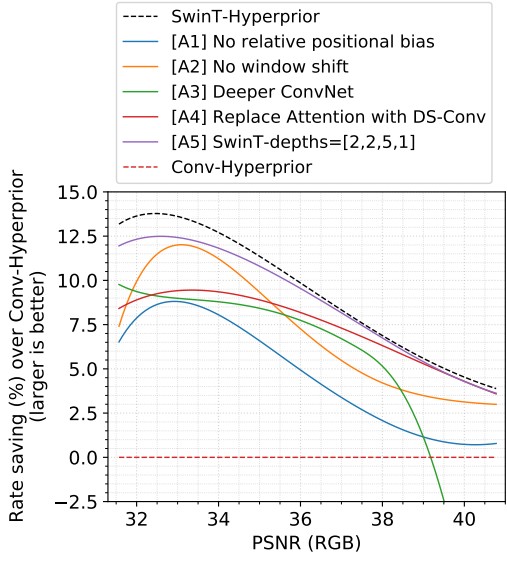

Figure 9: Ablation study

## 5 CONCLUSION

In this work we propose Swin transformer based transforms for image and video compression. In the image compression setting, SwinT transform consistently outperforms its convolutional counterpart. Particularly, the proposed SwinT-ChARM model outperforms VTM-12.1 at comparable decoding speed, which, to the best of our knowledge, is the first in learning-based methods. We also show the effectiveness of SwinT transforms when extended to the P-frame compression setting. Compared with convolution transforms, SwinT transforms can spatially decorrelate the latent better, have more flexible receptive field to adapt to tasks that requires either short-range (image) and long-range (motion) information, and better progressive decoding of latent pixels. While pushing the neural image compression to a new level in terms of rate-distortion-computation trade-off, we believe it is only the starting point for developing more efficient transformer-based image and video codecs.

ACKNOWLEDGMENTS

We would like to thank Amir Said for developing entropy coding and great advice on data compression in general. We would also appreciate the helpful discussions from Reza Pourreza and Hoang Le, and draft reviews from Auke Wiggers and Johann Brehmer.

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

# Appendix

## Table of Contents

## A  MODELS

### A.1  CONVOLUTION BASELINES

**Conv-Hyperprior and Conv-ChARM** The architecture of Conv-Hyperprior and Conv-ChARM are shown in Figure 10 and Figure 11. For both architecture, our base model (i.e. medium size) has the following hyperparameters: $(C_1, C_2, C_3, C_4, C_5, C_6, C_7) = (320, 320, 320, 320, 192, 192, 192)$.

### A.2  SWIN-TRANSFORMER BASED COMPRESSION MODELS

**SwinT-Hyperprior, SwinT-ChARM** For both SwinT-Hyperprior and SwinT-ChARM, we use the same configurations: $(w_g, w_h) = (8, 4), (C_1, C_2, C_3, C_4, C_5, C_6) = (128, 192, 256, 320, 192, 192)$, $(d_1, d_2, d_3, d_4, d_5, d_6) = (2, 2, 6, 2, 5, 1)$ where $C$, $d$, and $w$ are defined in Figure 13 and Figure 2. The head dim is 32 for all attention layers in SwinT-based models.

**SwinT-SSF** For SwinT transforms used in SSF variant, the first Patch Merge block is with downsample rate of 4 and two other Patch Merge blocks with downsampling rate of 2. Thus the downsampling rate for the encoder is still 16, the same as the image compression models. There are only 3 transformer stages with depths 2, 4, 2. The embedding dim is 96. The number of latent and hyper latent channels are all 192. The window size is 4 for flow codec and 8 for residual codec.

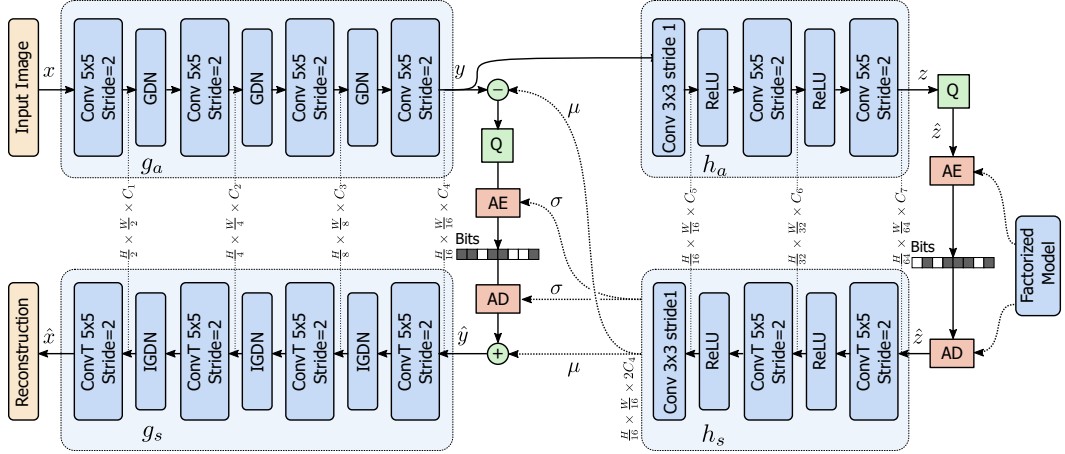

Figure 10: Conv-Hyperprior

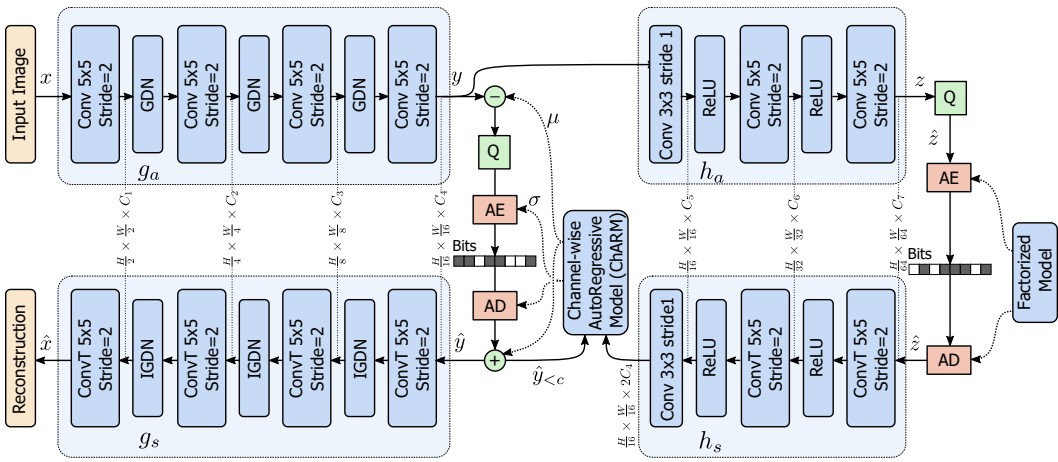

Figure 11: Conv-ChARM

**SwinT-SSF-Res**   This is a variant where only residual autoencoder uses SwinT transforms. Same architecture as the residual autoencoder in SwinT-SSF.

### A.3    MODEL CONFIGURATIONS FOR MODEL SIZE SCALING STUDY

#### A.3.1    SWINT-HYPERPRIOR

Set of model hyperparameters that are common to all experiments: $(d_1, d_2, d_3, d_4, d_5, d_6) = (2, 2, 6, 2, 5, 1)$ $(w_g, w_h) = (8, 4)$

**SwinT-Hyperprior (small)**   $(C_1, C_2, C_3, C_4, C_5, C_6) = (96, 128, 160, 192, 96, 128)$

**SwinT-Hyperprior (medium)**   $(C_1, C_2, C_3, C_4, C_5, C_6) = (128, 192, 256, 320, 192, 192)$

**SwinT-Hyperprior (large)**   $(C_1, C_2, C_3, C_4, C_5, C_6) = (160, 256, 352, 448, 192, 256)$

#### A.3.2    CONV-HYPERPRIOR

**Conv-Hyperprior (small)**   $(C_1, C_2, C_3, C_4, C_5, C_6, C_7) = (192, 192, 192, 192, 128, 128, 128)$

**Conv-Hyperprior (medium)**   $(C_1, C_2, C_3, C_4, C_5, C_6, C_7) = (320, 320, 320, 320, 192, 192, 192)$

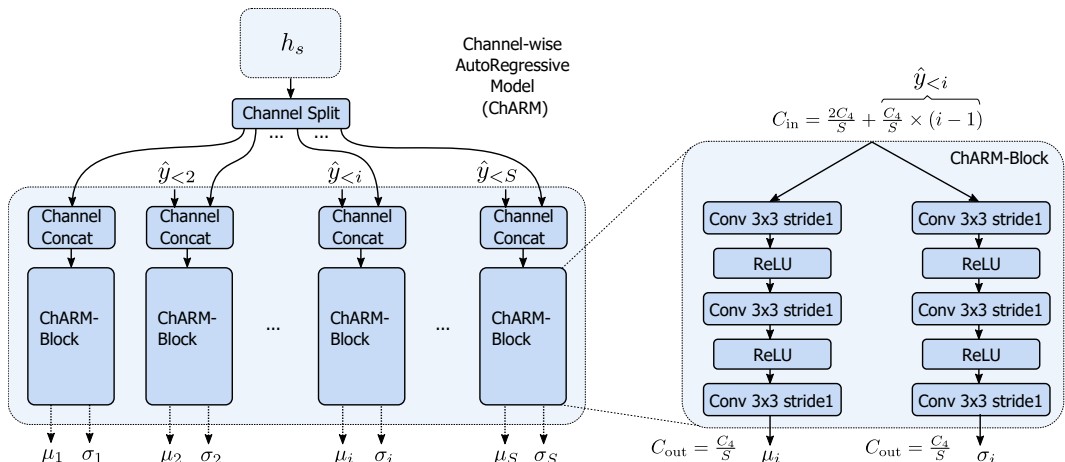

Figure 12: ChARM architecture. Since $y_i$ can only be decoded after $\mu_i$ and $\sigma_i$ is obtained, the $S$ ChARM-blocks are executed sequentially. We use $S = 10$ in all our experiments, which is consistent with (Minnen & Singh, 2020).

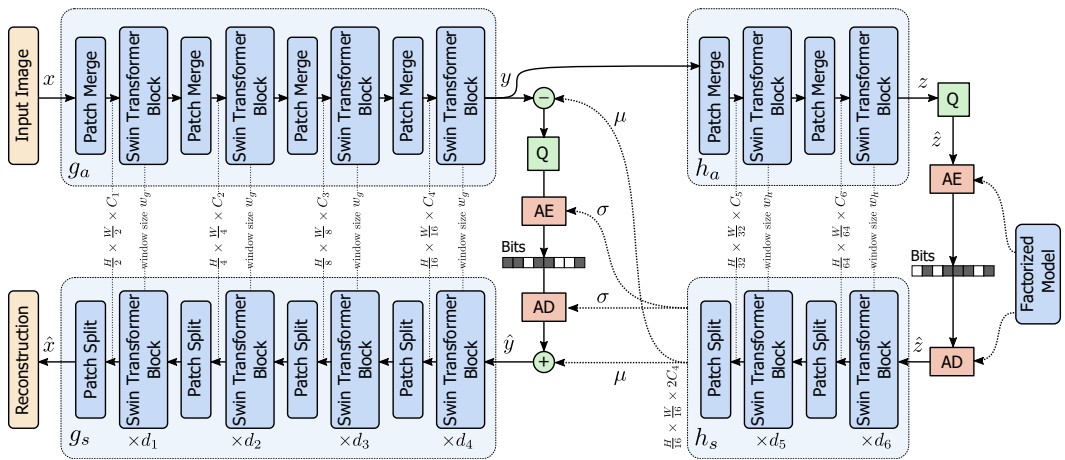

Figure 13: SwinT-Hyperprior

**Conv-Hyperprior (large)** $(C_1, C_2, C_3, C_4, C_5, C_6, C_7) = (448, 448, 448, 448, 256, 256, 256)$

# B TRAINING AND EVALUATION

## B.1 TRAINING

All image compression models are trained on CLIC2020 training set, which contains both professional and mobile training sets, in total 1,633 high resolution natural images. Conv-Hyperprior and SwinT-Hyperprior are trained with 2M batches. Each batch contains 8 patches of size $256 \times 256$ randomly cropped from the training images. Learning rate starts at $10^{-4}$ and is reduced to $10^{-5}$ at 1.8M step.

For Conv-ChARM, we first train a model Conv-ChARM at $\beta = 0.0001$ from scratch for 2M steps, with it as the starting point, we continue to train other beta values Conv-ChARM-$\beta$, $\beta \in B$ for 1.5M steps. For SwinT-ChARM-$\beta$, we load the transform weights from the checkpoint at 2M step of the pretrained SwinT-Hyperprior-$\beta$, then finetune the transforms together with the random initialized ChARM prior for 1.1M steps. Learning rate starts at $10^{-4}$ and is reduced to $10^{-5}$ for the last 100K steps.

Training loss $L = D + \beta R$ is a weighted combination of distortion $D$ and bitrate $R$, with $\beta$ being the Lagrangian multiplier steering rate-distortion trade-off. Distortion $D$ is MSE in RGB color space. To cover a wide range of rate and distortion, for each solution, we train 5 models with $\beta \in B = \{0.003, 0.001, 0.0003, 0.0001, 0.00003\}$.

Usually we need to train longer for the model with larger bitrates (i.e. smaller $\beta$) to converge. Particularly for the results presented in this paper, We train SwinT-Hyperprior-0.00003 for 2.5M steps instead of 2M steps for the other 4 lower bitrates.

For P-frame compression models, we follow the training setup of SSF. Both Conv-SSF and SwinT-SSF are trained on Vimeo-90k Dataset (Xue et al., 2019) for 1M steps with learning rate $10^{-4}$, batch size of 8, crop size of $256 \times 256$, followed by 50K steps of training with learning rate $10^{-5}$ and crop size[12] $384 \times 256$. The models are trained with 8 $\beta$ values $2^{\gamma} \times 10^{-4} : \gamma \in \{0, 1, ..., 7\}$. We adopt one critical trick to stablize the training from (Jaegle et al., 2021; Meister et al., 2018), i.e. to forward each video sequence twice during one optimization step (mini-batch), once in the original frame order, once in the reversed frame order. When this trick is used, we set the batch size to be 4 instead of 8. Finally we add flow loss only between 0 and 200K steps, which we found not critical for stable training but helps improve the RD.

For all model training, Adam optimizer is used without weighted decay. Training for 2M steps takes about 10 days and 14 days respectively for Conv-Hyperprior and SwinT-Hyperprior on a single Nvidia V100 GPU. Total training time is about 7.5 days on a single Nvidia V100 GPU.

For all models, we use *mixed quantization* during training (Minnen & Singh, 2020), i.e. adding uniform noise to the continuous latent before passing to the prior model, subtracting prior mean from the continuous latent followed by rounding before passing to the decoder transform.

---

[12]We did not use the crop size $384 \times 384$ during the second stage as in the original paper because the resolution of Vimeo dataset is $448 \times 256$. We found in our case increasing crop size from $256 \times 256$ to $384 \times 256$ in the second stage does not improve RD.

## B.2 TRADITIONAL CODEC EVALUATION

In this section, we provide evaluation script used to generate results for traditional codecs.

### B.2.1 IMAGE CODECS

**VTM-12.1:** VTM-12.1 software is built from `https://vcgit.hhi.fraunhofer.de/jvet/VVCSoftware_VTM/-/tags/VTM-12.1` and we use the script from CompressAI (`https://github.com/InterDigitalInc/CompressAI/tree/efc69ea24`) for dataset evaluation. Specifically, the following command is issued to gather VTM-12.1 image compression evaluation results:

```
python -m compressai.utils.bench vtm [path to image folder]
-c [path to VVCSoftware_VTM folder]/cfg/encoder_intra_vtm.cfg
-b [path to VVCSoftware_VTM folder]/bin
-q 16, 18, 20, 22, 24, 26, 28, 30, 32, 34, 36, 38, 40
```

**BPG:** BPG software is obtained from `https://bellard.org/bpg/` and the following commands are used for encoding and decoding.

```
bpgenc -e x265 -q [0 to 51] -f 444
-o [encoded heic file] [original png file]
bpgdec -o [decoded png file] [encoded heic file]
```

### B.2.2 VIDEO CODECS

**HEVC (x265)**

```
ffmpeg -y -pix_fmt yuv420p -s [resolution] -r [frame-rate]
-i [input yuv420 raw video] -c:v libx265 -preset medium
-crf [9, 12, 15, 18, 21, 24, 27, 30] -tune zerolatency
-x265-params "keyint=12:min-keyint=12:verbose=1"
[output mkv file path]
```

**AVC (x264)**

```
ffmpeg -y -pix_fmt yuv420p -s [resolution] -r [frame-rate]
-i [input yuv420 raw video] -c:v libx264 -preset medium
-crf [9, 12, 15, 18, 21, 24, 27, 30] -tune zerolatency
-x264-params "keyint=12:min-keyint=12:verbose=1"
[output mkv file path]
```

**HEVC (HM)**

```
[Path to HM folder]/bin/TAppEncoderStatic
-c [Path to HM folder]/cfg/encoder_lowdelay_P_main.cfg
-i [input yuv raw video] --InputBitDepth=8 -wdt [width]
-hgt [height] -fr [frame-rate] -f [number of frames]
-o [output yuv video] -b [encoded bitstream bin file]
-ip 12  -q [12, 17, 22, 27, 32, 37, 42]
```

## C  BD RATE COMPUTATION

```
def Bjontegaard_Delta_Rate(
        # rate and psnr in ascending order
        rate_ref, psnr_ref, # reference
        rate_new, psnr_new, # new result
    ):
    min_psnr = max(psnr_ref[0], psnr_new[0], 30)
```

```
max_psnr = min(psnr_ref[-1], psnr_new[-1], 44)

log_rate_ref = log(rate_ref)
log_rate_new = log(rate_new)

spline_ref = scipy.interpolate.CubicSpline(
    psnr_ref, log_rate_ref, bc_type='not-a-knot',
    extrapolate=True,
)
spline_new = scipy.interpolate.CubicSpline(
    psnr_new, log_rate_new, bc_type='not-a-knot',
    extrapolate=True,
)

delta_log_rate = (
    spline_new.integrate(min_psnr, max_psnr) -
    spline_ref.integrate(min_psnr, max_psnr)
)

delta_rate = exp(delta_log_rate / (max_psnr - min_psnr))

return 100 * (delta_rate - 1)
```

## C.1  BD RATE FOR IMAGE CODEC

```
# Evaluate BD-rate on an image dataset
bd_rates = list()
for image in image_dataset:
    # evaluate rate and psnr on reference and new codec
    # for this image with different qualities
    rate_ref, psnr_ref = ReferenceCodec(image, qp=[...])
    rate_new, psnr_new = NewImageCodec(image, beta=[...])
    bd_rates.append(
        Bjontegaard_Delta_Rate(
            rate_ref, psnr_ref,
            rate_new, psnr_new,
        )
    )
# BD is computed per image and then averaged
bd_rate = bd_rates.mean()
```

## C.2  BD RATE FOR VIDEO CODEC

```
# Evaluate BD-rate on a video dataset
bd_rates = list()
for video in video_dataset:
    # evaluate rate and psnr on reference and new codec
    # for this video with different qualities
    rate_ref, psnr_ref = ReferenceCodec(video, qp=[...])
    rate_new, psnr_new = NewVideoCodec(video, beta=[...])
    bd_rates.append(
        Bjontegaard_Delta_Rate(
            rate_ref, psnr_ref,
            rate_new, psnr_new,
        )
    )
# BD is computed per video and then averaged
bd_rate = bd_rates.mean()
```

# D MORE RESULTS

## D.1 IMAGE COMPRESSION

Additional rate-distortion results on CLIC2021, Tecnick, and JPEG-AI are provided in Figure 14, Figure 15, and Figure 16.

For a complete comparison of results from existing literatures, we provide a summary RD plot of all neural image codec solutions known to us in Figure 28 on Kodak. In Figure 29 and Figure 30, we plot the percentage rate saving with BPG444 and VTM-12.1 as reference, respectively.

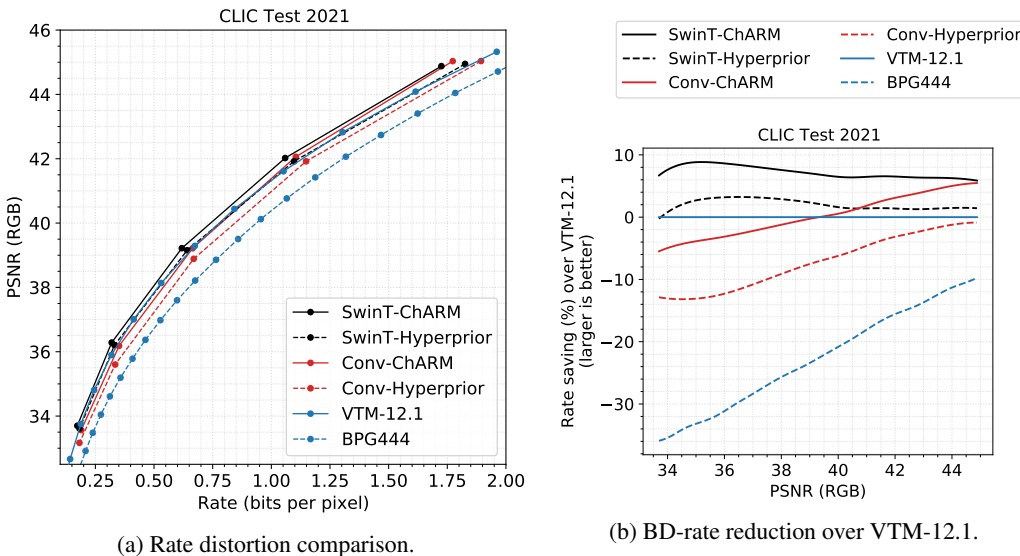

Figure 14: Comparison of compression efficiency on CLIC Test 2021.

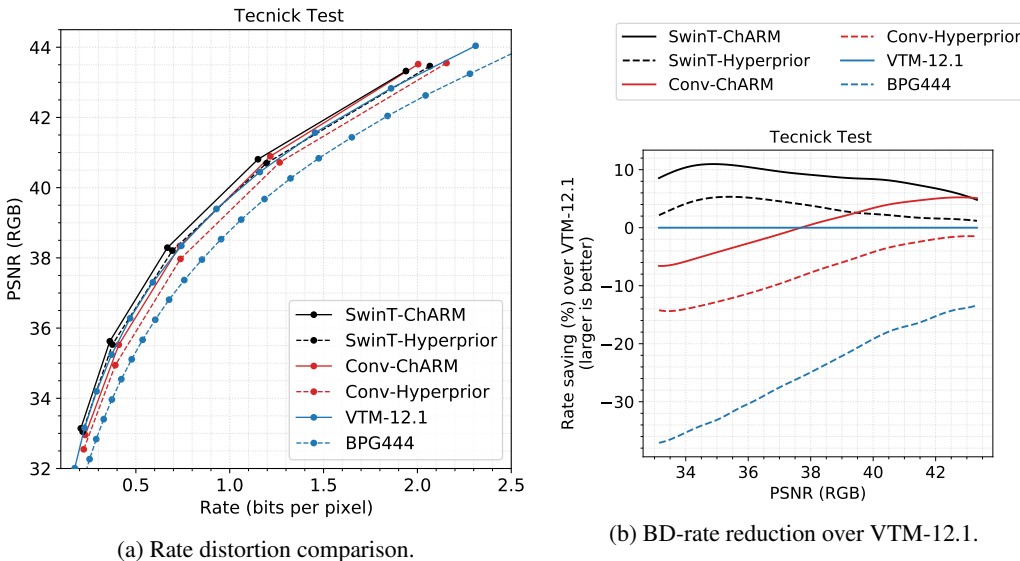

Figure 15: Comparison of compression efficiency on Tecnick Test.

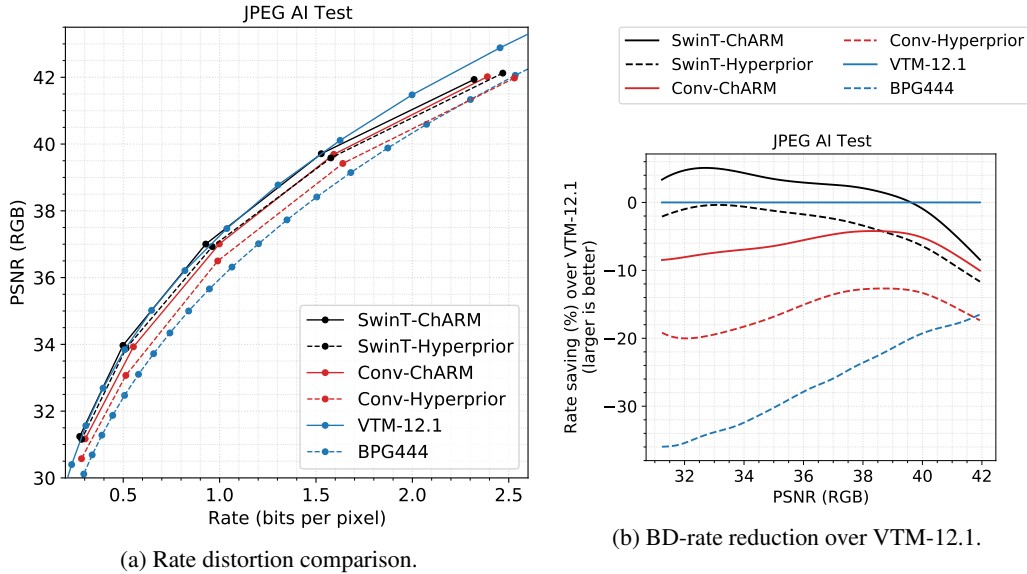

(a) Rate distortion comparison.

(b) BD-rate reduction over VTM-12.1.

Figure 16: Comparison of compression efficiency on JPEG AI Test.

## D.2 VIDEO COMPRESSION

In Figure 17 and Figure 18, we provide performance comparison of Conv-SSF, SwinT-SSF, HEVC (x265), and AVC (x264) with per-video breakdown.

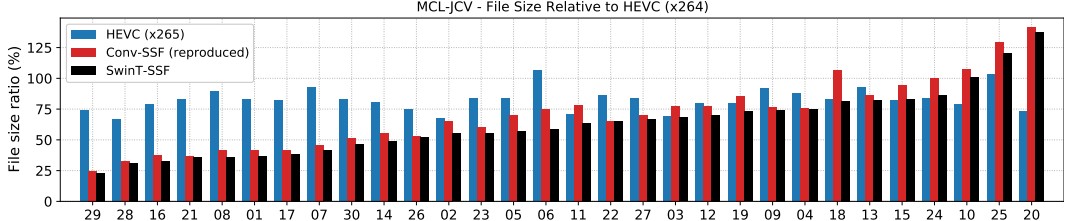

Figure 17: Rate savings for each video in the MCL-JCV dataset. Values represent the file size relative to H.264 as estimated by BD rate. [18, 20, 24, 25] are animated sequences.

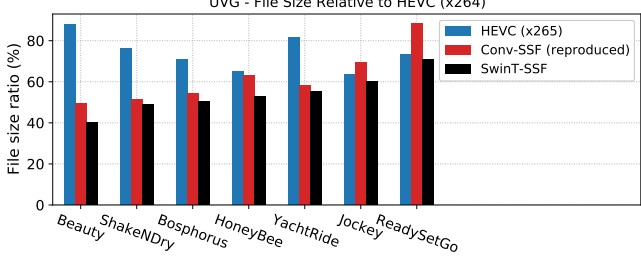

Figure 18: Rate savings for each video in the UVG dataset. Values represent the file size relative to H.264 as estimated by BD rate.

## D.3 CODING COMPLEXITY

We evaluate the decoding complexity of all neural image codecs in terms of the time for network inference and entropy coding, peak GPU memory, model size, etc. We select 100 high resolution images from CLIC testset and center crop them to three resolutions ($768 \times 512, 1280 \times 768, 1792 \times$

1024) to see how those metrics scale with image size. Batch size is one for all model inference. We run the experiment on a local workstation with one RTX 2080 Ti GPU, with PyTorch 1.9.0 and Cuda toolkit 11.1. For bit-exact entropy coding, we need to use deterministic[13] convolution. The neural networks run on the single GPU and entropy coding runs on CPU with 8 threads. We follow the standard protocols to measure the inference time and peak memory of neural nets, such as GPU warm up and synchronization. File open/close is excluded from coding time measurement. MACs, GPU peak memory, and model parameter count are profiled using `get_model_profile` function of deepspeed profiler[14].

We show more details on the coding complexity of neural image codecs in Figure 19, particularly the linear scaling to image resolution of both SwinT-based and Conv-based models. The break-down of encoding complexity is shown in Table 4.

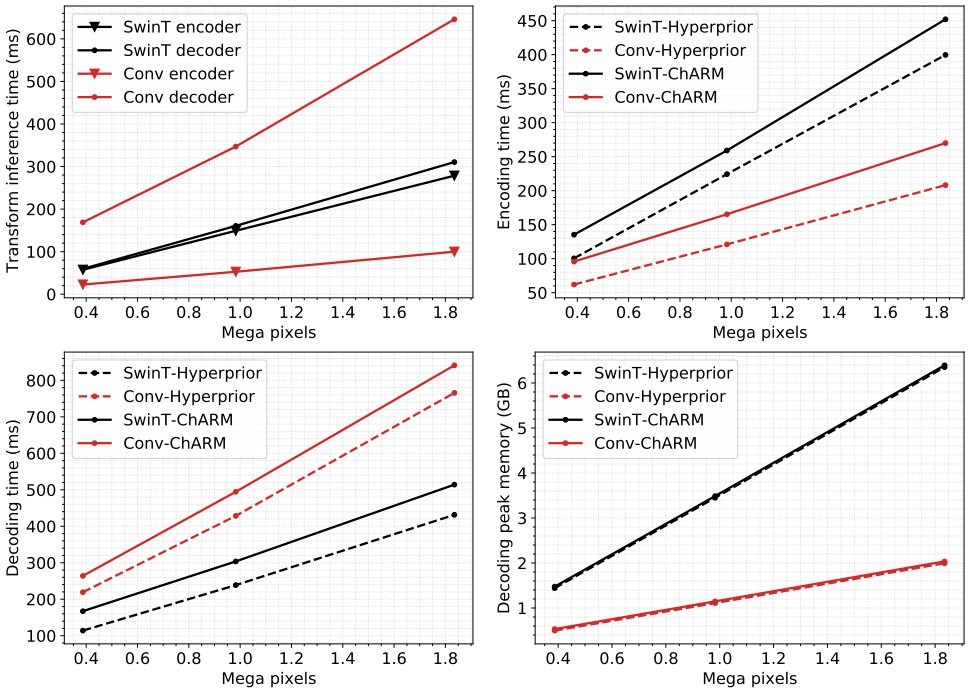

Figure 19: Detailed profiling on a single RTX 2080 Ti with deterministic conv. Note that while Conv encoder and Conv decoder are of symmetric architectures, Conv decoder takes more time than Conv encoder mainly because Transposed Conv layers in the decoder take much longer than Conv layers in the encoder.

For completeness, we also report the profiling for CPU coding time in Table 5 and Table 6. The evaluation setup is the same as the GPU profiling case, except models are run on the CPU instead (same host machine with Intel(R) Xeon(R) W-2123 CPU @ 3.60GHz).

Table 7 reports encoding and decoding time of VTM-12.1 under different quantization parameters (QPs), evaluated on an Intel Core i9-9940 CPU @ 3.30GHz, averaged over 24 Kodak images. As can be seen from the table, decoding time of VTM-12.1 is a function of reconstruction quality, where longer decoding time is observed for higher quality reconstruction. In Figure 1, the reported VTM-12.1 decoding speed corresponds to a QP value of 28, where the bpp value is similar to that obtained by models trained with $\beta = 0.001$. It is worth pointing out that VTM-12.1 encoding process is much slower, ranging anywhere from 1 to 5 minutes per image, whereas neural codec runs much faster.

---

[13]https://pytorch.org/docs/1.9.0/generated/torch.use_deterministic_algorithms.html?highlight=deterministic#torch.use_deterministic_algorithms

[14]https://www.deepspeed.ai/tutorials/flops-profiler/#usage-outside-the-deepspeed-runtime

Table 4: GPU encoding complexity. All models are trained with $\beta = 0.001$, evaluated on the resolution $768 \times 512$ (the average bitrate is around 0.7 bpp). For the encoding time, we show the network inference time for the encoder $g_a$, the hyper encoder $h_a$, the hyper decoder $h_s$, the entropy encoding time for the hyper latent $\hat{z}$ and the latent $\hat{y}$ (including network inference time for the prior model if it is ChARM). GMACs and GPU peak memory during encoding and the parameter count of the entire model are also listed.

| Codec | Time (ms) | | | | | | GMACs | Memory (GB) | Params (M) |
|---|---|---|---|---|---|---|---|---|---|
| | $g_a$ | $h_a$ | $\hat{z}$ | $h_s$ | $\hat{y}$ | total | | | |
| Conv-Hyperprior | 22.7 | 2.3 | 2.8 | 4.0 | 29.8 | 62.0 | 102 | 0.62 | 21.4 |
| SwinT-Hyperprior | 57.5 | 1.2 | 2.4 | 4.8 | 34.3 | 100.5 | 100 | 1.47 | 24.7 |
| Conv-ChARM | 22.9 | 2.4 | 2.6 | 4.1 | 63.4 | 95.7 | 114 | 0.66 | 29.3 |
| SwinT-ChARM | 57.8 | 1.3 | 2.4 | 4.8 | 68.7 | 135.3 | 112 | 1.51 | 32.6 |

Table 5: CPU decoding complexity. All models are trained with $\beta = 0.001$, evaluated on $768 \times 512$ images (average 0.7 bpp). Decoding time is broken down into inference time of hyper-decoder $h_s$ and decoder $g_s$, entropy decoding time of hyper-code $\hat{z}$ and code $\hat{y}$ (including inference time of the prior model if it is ChARM).

| Codec | CPU decoding Time (ms) | | | | | GMACs |
|---|---|---|---|---|---|---|
| | $\hat{z}$ | $h_s$ | $\hat{y}$ | $g_s$ | total | |
| Conv-Hyperprior | 5.1 | 20.9 | 30.5 | 1636.2 | 1703.3 | 350 |
| SwinT-Hyperprior | 5.7 | 18.3 | 30.3 | 1912.8 | 1979.5 | 99 |
| Conv-ChARM | 4.7 | 20.9 | 154.6 | 1634.3 | 1825.6 | 362 |
| SwinT-ChARM | 5.8 | 18.2 | 141.1 | 1877.3 | 2054.7 | 111 |

Table 6: CPU encoding complexity. All models are trained with $\beta = 0.001$, evaluated on the resolution $768 \times 512$ (the average bitrate is around 0.7 bpp). For the encoding time, we show the network inference time for the encoder $g_a$, the hyper encoder $h_a$, the hyper decoder $h_s$, the entropy encoding time for the hyper latent $\hat{z}$ and the latent $\hat{y}$ (including network inference time for the prior model if it is ChARM).

| Codec | CPU encoding Time (ms) | | | | | | GMACs |
|---|---|---|---|---|---|---|---|
| | $g_a$ | $h_a$ | $\hat{z}$ | $h_s$ | $\hat{y}$ | total | |
| Conv-Hyperprior | 896.0 | 8.9 | 2.7 | 20.9 | 31.1 | 960.9 | 102 |
| SwinT-Hyperprior | 1781.6 | 14.5 | 2.6 | 18.3 | 29.7 | 1847.9 | 100 |
| Conv-ChARM | 895.7 | 8.9 | 2.5 | 20.9 | 146.3 | 1075.5 | 114 |
| SwinT-ChARM | 1697.1 | 14.5 | 2.6 | 18.2 | 133.5 | 1867.1 | 112 |

Table 7: Decoding and encoding time of VTM-12.1 averaged over 24 Kodak images.

| QP | bits per pixel | PSNR (dB) | Decoding time(s) | Encoding time (s) |
|---|---|---|---|---|
| 16 | 2.5441 | 44.09 | 0.284 | 300.47 |
| 28 | 0.7844 | 36.76 | 0.249 | 118.85 |
| 40 | 0.1557 | 29.51 | 0.149 | 71.10 |

# E   ANALYSIS

## E.1   SPATIAL CORRELATION OF LATENT

We visualize the spatial correlation map for Conv-Hyperprior and SwinT-Hyperprior at different $\beta$ in Figure 20.

### E.2 Effective Receptive Field

See Figure 21 for the effective receptive field for the composed encoding tranforms $h_a \circ g_a$ and Figure 22 for the mean attention distance visualization of each head within each transformer layer.

### E.3 Rate distribution across latent channels

It is generally believed that ConvNets learn to extract various features and store them in each channel of the activations. Here we look into the features in the latent channels which are to be quantized and entropy coded to bitstreams. Particularly we order the total bitrate of each channel averaged over Kodak dataset (24 $768 \times 256$ images). The result is shown in Figure 23. We find an interesting phenomenon across models under different bitrates: there is a cutoff point of the bitrate-vs-channel curve where the bitrate suddenly drops to zero, which manifest the rate constraint in the loss function. As expected, the cutoff index decreases for the model trained for smaller bitrate (larger $\beta$).

### E.4 Centered kernel alignment

To investigate the difference or similarity between latent features of Conv-based and SwinT-based models, we resort to a commonly used tool in representation learning called centered kernel alignment (CKA) (Kornblith et al., 2019). We evaluate CKA between each of the Conv latent channel and SwinT latent channel (both models are trained under the same $\beta$) over the 24 Kodak images. There are 320 channels for both Conv latent and SwinT latent, resulting a $320 \times 320$ CKA matrix. The result is shown in Figure 24. The latent channel is ordered by the averaged bitrate of each channel over Kodak images (same as in Section E.3). The CKA matrix has clear block structure, where high similarity region corresponds to the latent channels before the bitrate cutoff in the rate distribution curve (Figure 23).

**Identification of SwinT and Conv latent channels with CKA**     Within the block of high similarity (from the CKA matrix), we identify the 'less' similar SwinT latent channels with lowest CKA values between this SwinT channel and all other Conv latent channels. For each of the identified SwinT channel, we find the Conv latent channel with the largest CKA value between the two. This way, we are able to identify latent channels of two different models with high similarity. We show the identified top 8 channels in Figure 25. The channels are indeed highly similar, up to a sign flip, even through the two model architectures are quite different. This empirical result is relevant to the literature on the identifiability of generative models (Khemakhem et al., 2020).

### E.5 Progressive decoding

More visual examples of reconstructions of checkerboard (spatially) masked latent are provided in Figure 26.

**Channel-wise progressive decoding**     Here we visualize the behavior of channel-wise progressive decoding of Conv and SwinT models. For both models, we order the latent channels with a heuristic importance metric: the reduction of distortion over the increase of rate if one channel is included for decoding. We start with the order of bitrate per channel, we pass the leading channels of bitrate order (zero out all rest channels) to the decoder to obtain reconstruction and calculate distortion. We plot the top 8 channels following this importance order. For each channel, we show 6 maps from top to bottom: latent values, mean prediction from the hyper decoder, standard deviation prediction from the hyper decoder, the bitmap, the reconstruction with only current channel, the reconstruction with up to current channel (all leading channels). The result is shown in Figure 27. For Conv models, usually the top 3 important channels are responsible for lightness, and two color components (blue-yellow, red-green), similar to the LAB colorspace. The rest of the latent channels are responsible for adding details like texture and edges. For the Swint models, at low bitrate, there is one significantly different channel (the first column in Figure 27b), which is in a coarse scale (with smooth blocks) and responsible for the reconstruction of a nearly constant image with value close to 120 (the mean value of natural image dataset). This latent channel costs extremely small bitrate but reaches PSNR of 13dB. We tried remove this first channel, the progressive reconstruction with the rest leading 7 channels only leads to PSNR around 16dB, instead of 26dB shown in the figure.

## F    MORE ABLATION STUDIES

**Local self-attention**    To see if local self-attention is the most important component in transformers, we replace it by depthwise separable convolution block[15] (Han et al., 2021; El-Nouby et al., 2021), which performs similar spatial feature aggregation as self-attention, while keeping all other components the same as in the SwinT. We found this change only leads to minor degradation in RD. This suggests other components in transformers such as MLPs and skip connections may also play a big role, other than just self-attention, for the leading performance in our work and many other tasks (Dong et al., 2021).

**Small depths**    Upon investigating the mean attention distance as shown in Figure 22, we find the last block in each of the last two encoder stages has about half of its attention heads degenerate to attending to fixed nearly pixels. This suggests redundant transformer blocks at that stage, so we remove those two blocks, i.e. from depths $[2, 2, 6, 2]$ to $[2, 2, 5, 1]$. The resulting SwinT-Hyperprior has even less parameters (20.6M) than Conv-Hyperprior (21.4M) while with almost no RD loss compared to the larger model. We expect more hyperparameter search will identify models with better RD-complexity trade-off than we currently show in this work.

**Deeper Conv encoder**    Deeper models are usually more expressive (Raghu et al., 2017) and the state-of-the-art Conv-based compression models typically use much deeper layers than the encoder in the original Hyperprior model (Ballé et al., 2018). As a sanity check on whether deeper convolutional transforms can outperform SwinT-based encoder transforms with 12 blocks, we take an existing design (Chen et al., 2021) with residual blocks and attention (sigmoid gating) layers, which has over 50 conv layers in either encoder or decoder, and more parameters than conv baseline. It indeed improves the RD in lower bitrate, but still worse than SwinT-Hyperprior, and gets much worse in higher bitrate. This is probably the reason that compression models based on this type of transforms did not report results at higher bitrates.

---

[15]Conv1×1-LayerNorm-ReLU-DSConv3×3-LayerNorm-ReLU-Conv1×1-LayerNorm-ReLU

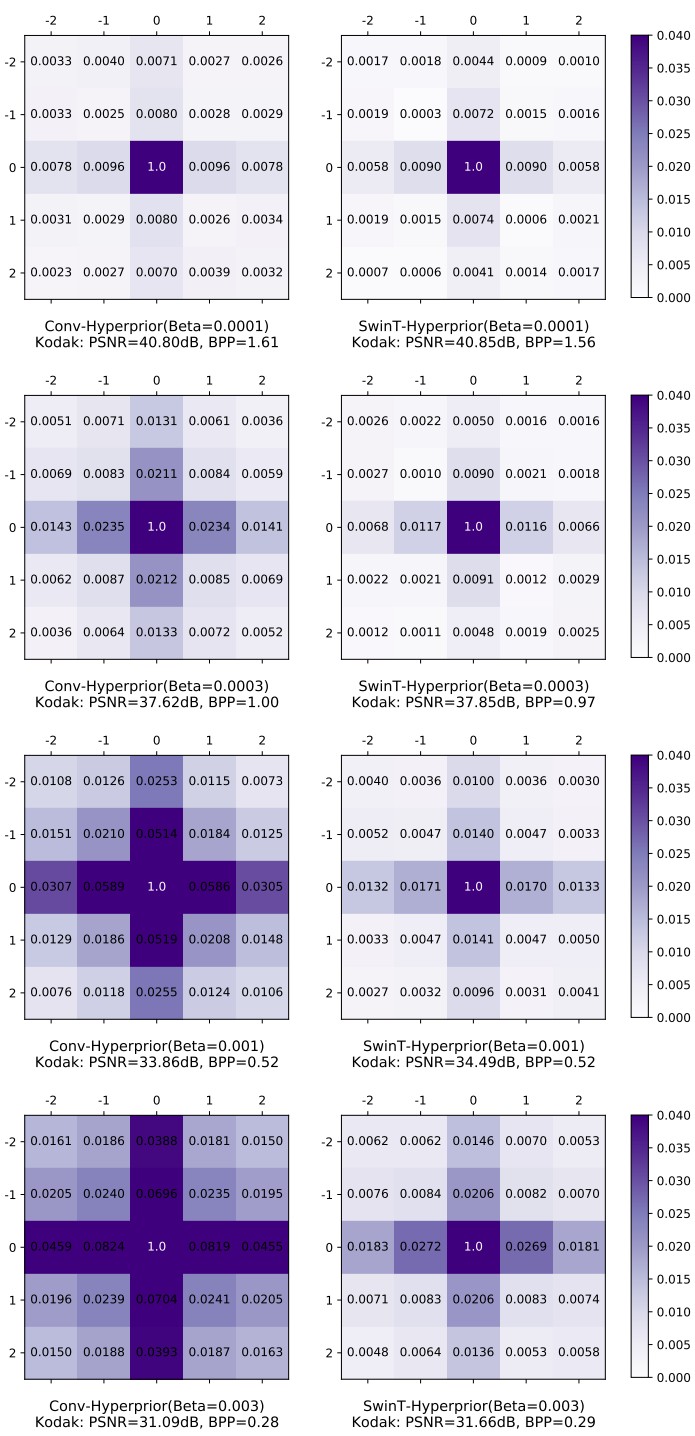

Figure 20: Spatial correlation of $(\mathbf{y} - \boldsymbol{\mu}(\hat{\mathbf{z}}))/\boldsymbol{\sigma}(\hat{\mathbf{z}})$, averaged across all latent elements of all images on Kodak. The value with index $(i, j)$ corresponds to the normalized cross-correlation of latents at spatial location $(w, h)$ and $(w + i, h + j)$. Left column corresponds to Conv-Hyperprior and right columns corresponds to SwinT-Hyperprior. Each row shows a pair of models trained at the same $\beta$, where the $\beta$ values from top to bottom are 0.0001, 0.0003, 0.001, 0.003. A consistent observation across all these models is that SwinT-Hyperprior achieves uniformly smaller correlation than its convolutional counterpart. As $\beta$ gets smaller (rate becomes lower), the correlation in both models increase, with SwinT-Hyperprior increasing much slower than Conv-Hyperprior.

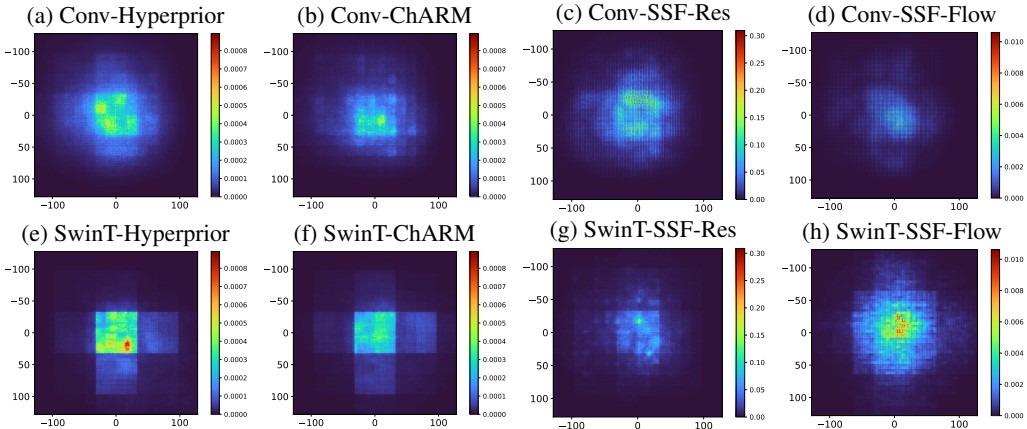

Figure 21: Comparison of effective receptive field (ERF) of the composed encoders $h_a \circ g_a$. The ERF is visualized as the absolution gradients of the center pixel in the hyper latent (i.e. $dz/dx$) with respect to the input images/videos, specifically 24 Kodak images cropped to $512 \times 512$ for image codecs (the left four models, all with $\beta = 0.003$), and 24 randomly selected batches from UVG video dataset cropped to $768 \times 768$ for P-frame codecs (the right four models, all with $\beta = 0.0008$).

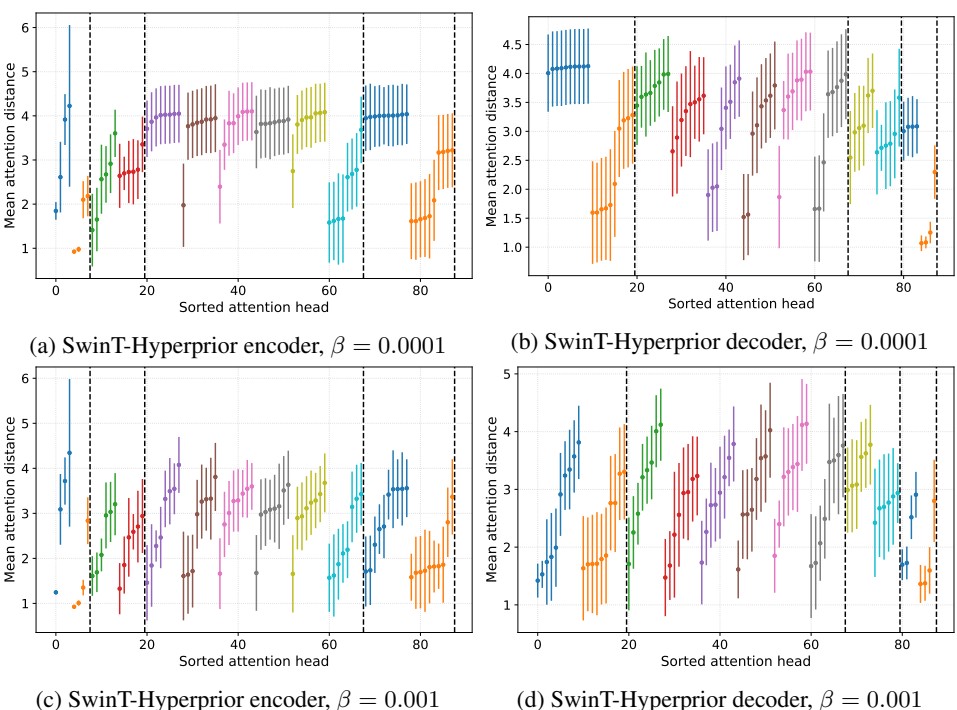

Figure 22: Mean attention distance of SwinT-Hyperprior models evaluated on Kodak. It is calculated as the average relative distance between each query and key weighted by the attention weight for each query, in each head of each layer. Each vertical color bar in the figure shows the mean and 1-std for the mean attention distance of one head, dashed black bar separates heads from different transformer stages (feature resolutions). The order of input to output is from left to right within each figure. Lower bitrate models have smaller attention distance.

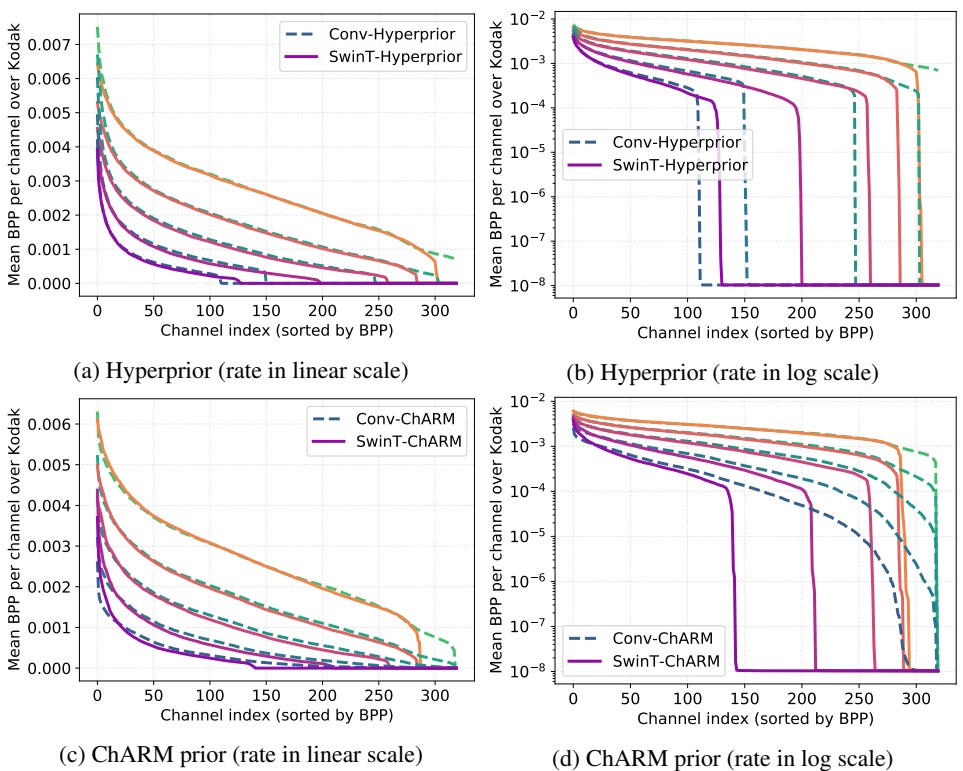

Figure 23: Rate distribution of latent channels. Five lines with the same colormap from bright to dark corresponds to the model trained under five $\beta$ values from small to large (i.e. overall bitrate from large to small). The cutoff index decreases as the total rate goes down.

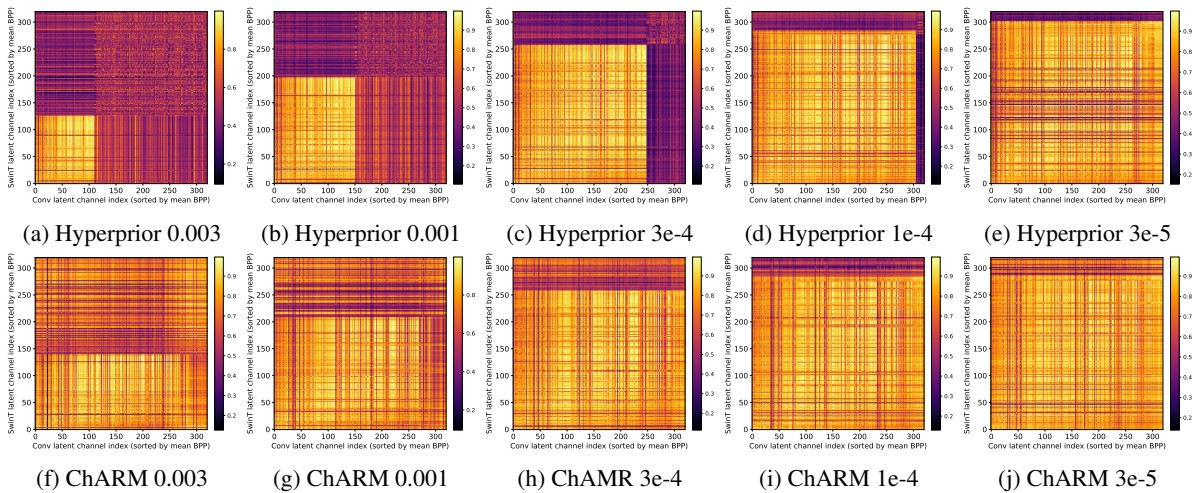

Figure 24: Centered Kernel Alignment between 320 SwinT latent channels and 320 Conv latent channels. Blocks with high similarity correspond to latent channels before the cutoff in Figure 23.

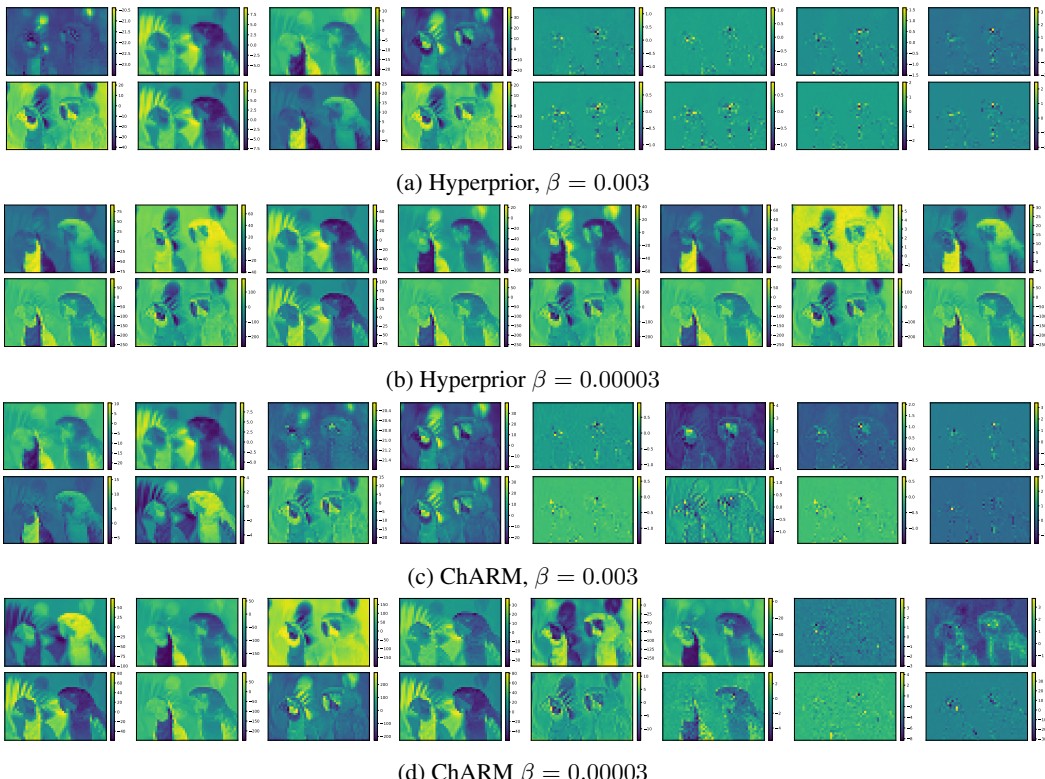

(a) Hyperprior, $\beta = 0.003$

(b) Hyperprior $\beta = 0.00003$

(c) ChARM, $\beta = 0.003$

(d) ChARM $\beta = 0.00003$

Figure 25: Identification of latent features from SwinT and Conv models with CKA (on 23rd image in Kodak). For each subfigure, top row shows 8 channels of SwinT latents with the smallest mean CKA values between this channel and all Conv latent channels (before the rate cutoff in Figure 23). The corresponding image on the bottom shows the Conv latent channel with the largest CKA with the top SwinT channel. The identified channels are highly similar, up to sign flip.

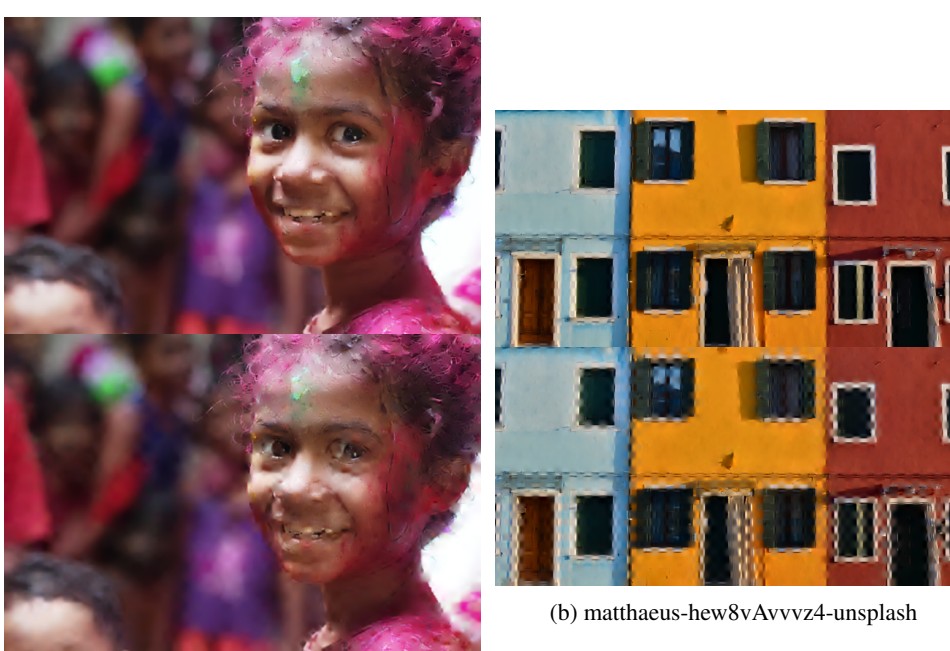

(a)    himanshu-singh-gurjar-iSi02D_Qx_w-
unsplash

(b) matthaeus-hew8vAvvvz4-unsplash

Figure 26: More examples of reconstructions with SwinT-Hyperprior (top row) and Conv-Hyperprior (bottom row) for the latent masked with checkerboard pattern where half of the latent dequantized values are replaced with the corresponding prior mean before passing to the decoder $g_s$. The example image shown in Figure 8b is andy-kelly-0E_vhMVqL9g-unsplash from CLIC2021 test, same as the two images shown here.

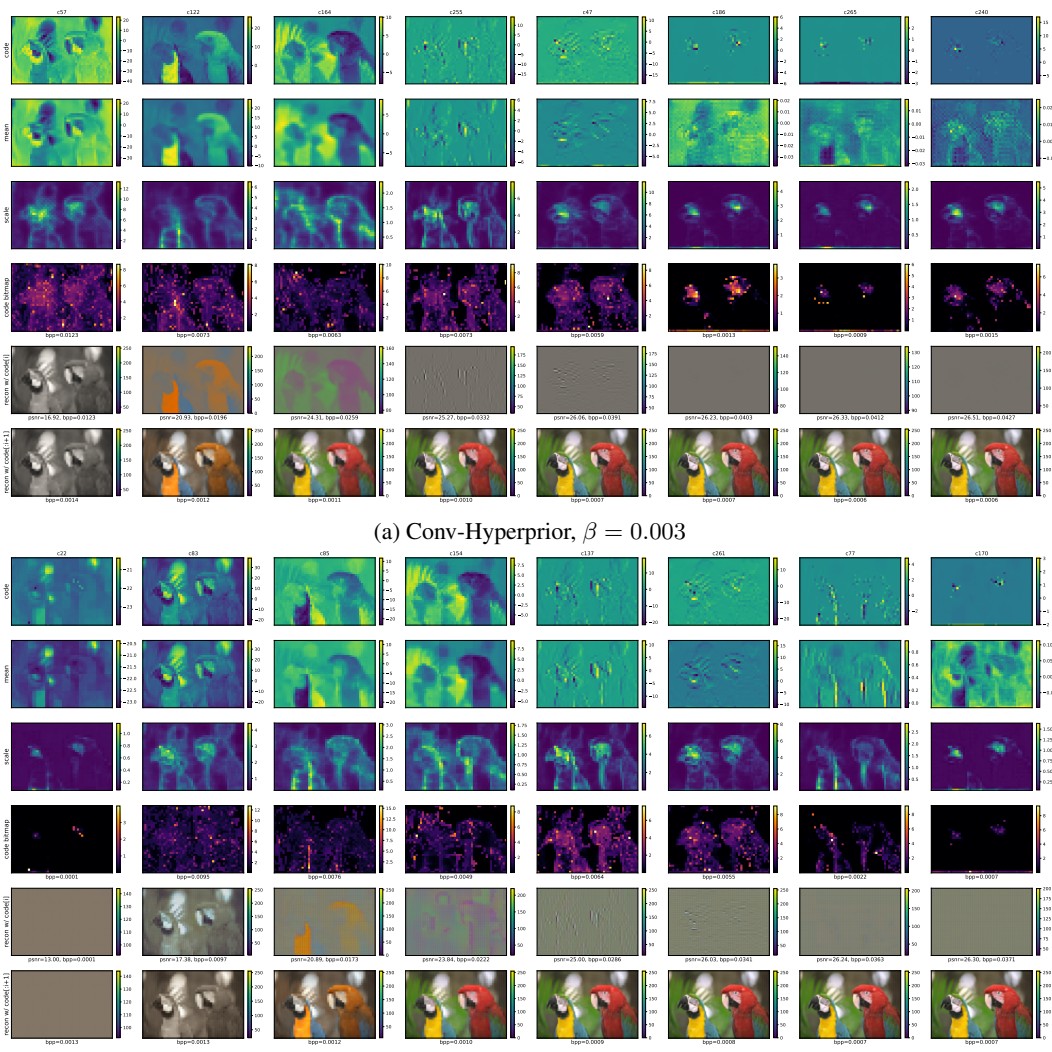

(a) Conv-Hyperprior, $\beta = 0.003$

(b) SwinT-Hyperprior $\beta = 0.003$

Figure 27: Channel-wise progressive decoding. The 3 leading latent channels following an importance order are responsible for grayscale, and two color components of the reconstruction (second to the last row). For SwinT models, we find a coarse-scale latent channel (the left most channel) which costs negligible bits but is important for reconstruction (without it, the 7 other leading channels can only decode to a reconstruction with PSNR around 16dB). For each subfigure, from top to bottom: latent, prior mean, prior std, bitmap, reconstruction with current channel, reconstruction with leading channels (up to current channel).

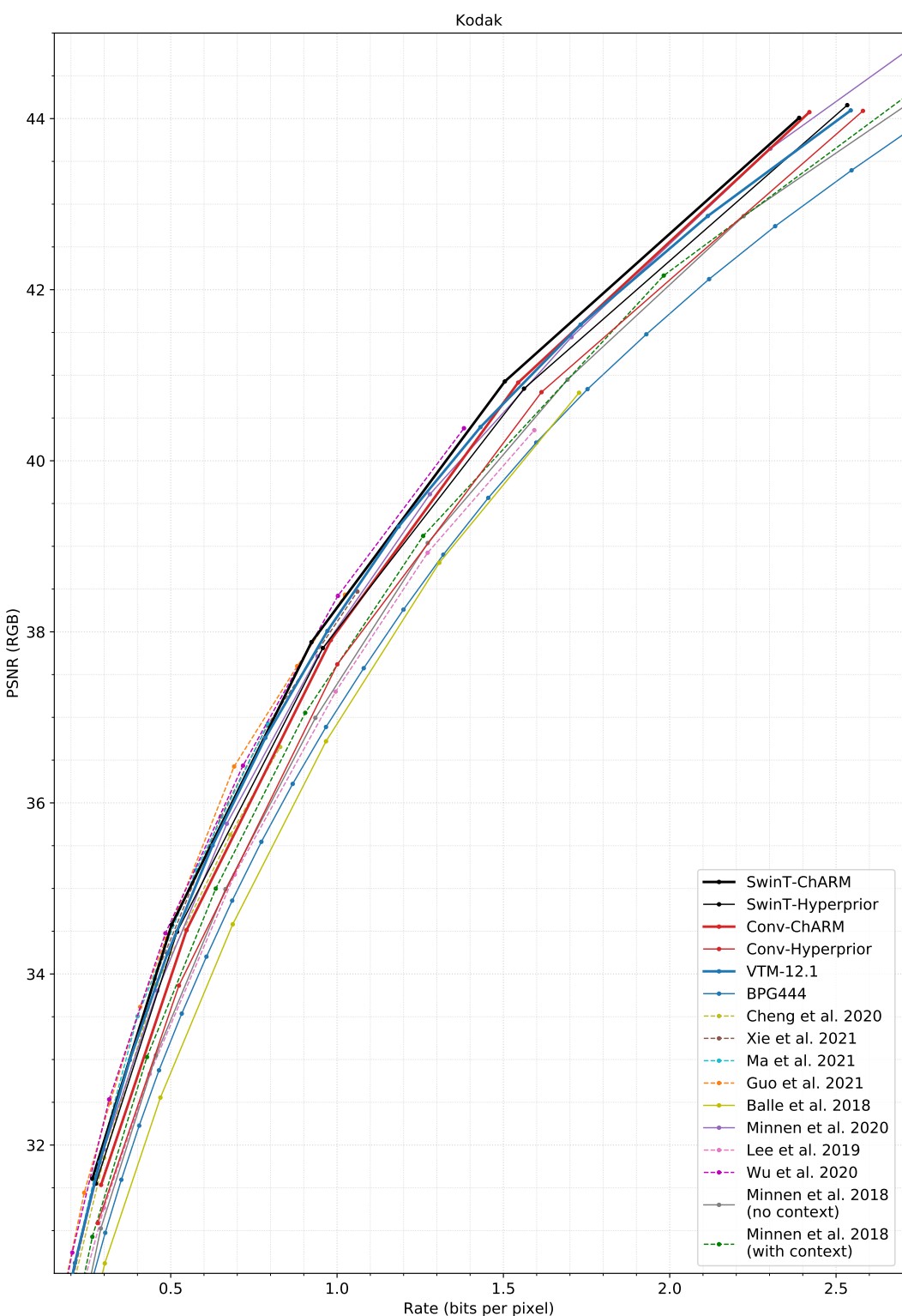

Figure 28: Rate-distortion performance on Kodak, comparing with existing works (Cheng et al., 2020; Xie et al., 2021; Ma et al., 2021; Guo et al., 2021; Ballé et al., 2018; Minnen & Singh, 2020; Lee et al., 2019; Wu et al., 2020; Minnen et al., 2018). Dashed line-style indicates the use of spatial autoregressive model (block-wise or pixel-wise) as prior model.

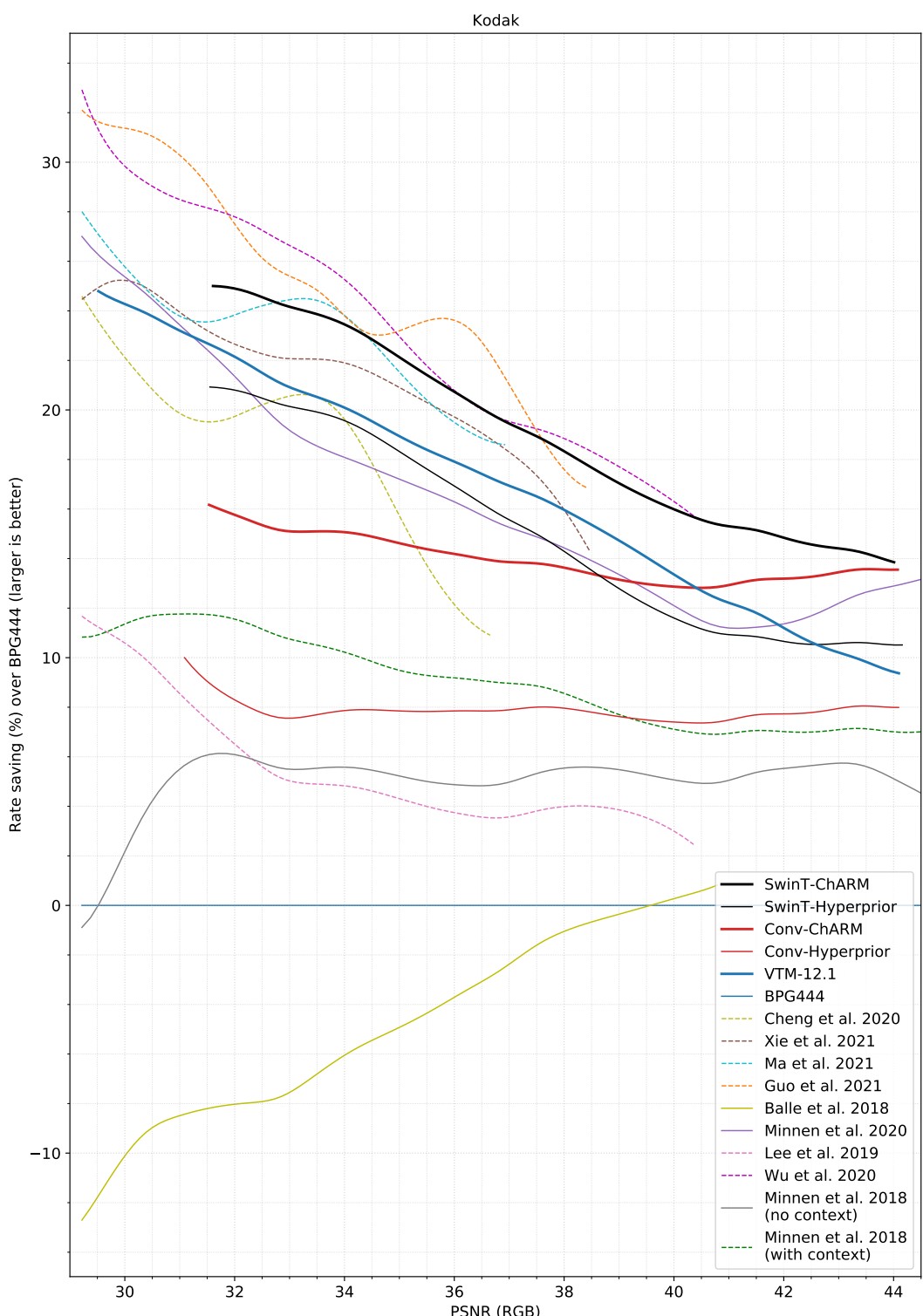

Figure 29: Percentage of rate-saving over BPG444 evaluated on Kodak (extended version of Figure 3b), comparing with existing works (Cheng et al., 2020; Xie et al., 2021; Ma et al., 2021; Guo et al., 2021; Ballé et al., 2018; Minnen & Singh, 2020; Lee et al., 2019; Wu et al., 2020; Minnen et al., 2018). Dashed line-style indicates the use of spatial autoregressive model (block-wise or pixel-wise) as prior model.

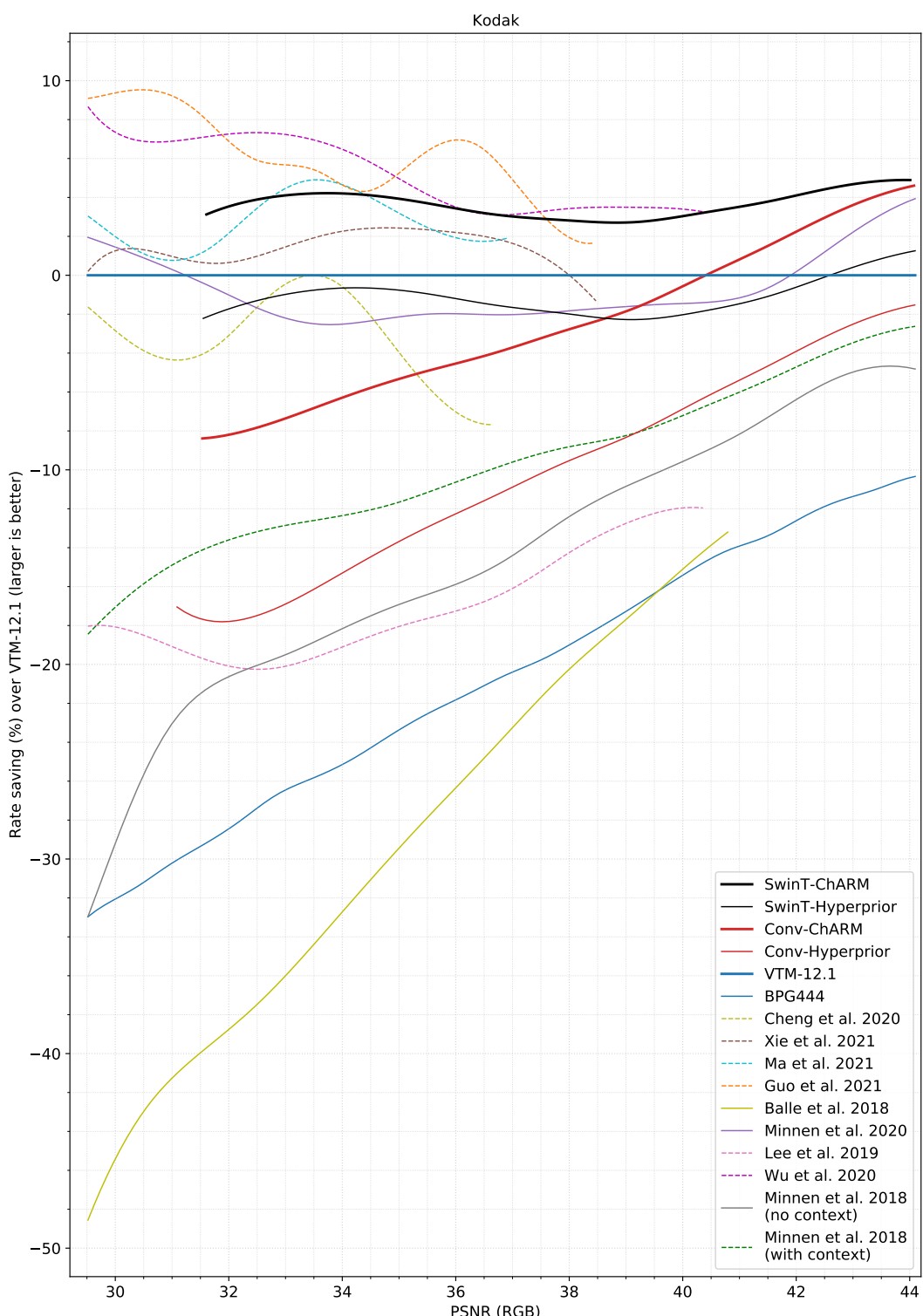

Figure 30: Percentage of rate-saving over VTM-12.1 evaluated on Kodak (extended version of Figure 3b), comparing with existing works (Cheng et al., 2020; Xie et al., 2021; Ma et al., 2021; Guo et al., 2021; Ballé et al., 2018; Minnen & Singh, 2020; Lee et al., 2019; Wu et al., 2020; Minnen et al., 2018). Dashed line-style indicates the use of spatial autoregressive model (block-wise or pixel-wise) as prior model.

