# OpenReview forum: "Transformer-based Transform Coding"
_ICLR.cc/2022/Conference — ICLR 2022 Poster_

### Official Review · Reviewer_U2rZ · 2021-10-30

**Correctness:** 4
**Technical Novelty And Significance:** 4
**Empirical Novelty And Significance:** 4
**Recommendation:** 8
**Confidence:** 5

**Main Review:**

Strengths:
The authors show strong results for SwinT networks across two entropy modeling networks (channel-wise autogressive and hyper prior) for image compression and scale-space architectures for video compression at much faster decoding times.

SwinT's high relative performance also appears to be more resilient to scaling the network to smaller capacities, as compared to a Conv model. The small SwinT is half the BD-rate loss of the Conv equivalent at under half the speed.

Latent analysis for SwinT shows a uniformly smaller correlation than the Conv models. It is a noteworthy result and leads to future research questions on if imposing that form leads to better image compression or if it is a result of a naturally better method. This question would also hold for the receptive field analysis, where SwinT's receptive fields are more rectilinear than the Conv models, which may be showing that Conv models are getting mostly unneeded latents influencing the reconstructions.

Overall, this paper includes an abundance of strong results, across multiple architectures and datasets for image and video compression. The appendices have a plethora of detail about training, architecture design and results that should make this paper very easy to reproduce and verify.


Weaknesses:
In Figure 19, SwinT models are shown to be much slower at encoding than Conv models while much faster at decoding. Is there a possible shift of compute going on in the SwinT networks that aren't being taken advantage of in the Conv models?

The gap at low bitrates is amazing, the SwinT models heavily out perform the Conv models, but that is gain disappears at higher quality / rates, any intuition on why that is?

The image and video codec baselines were used in medium and with low/zero latency optimizations enabled. This will generally put classic codecs at a lower R/D in comparisons. This would generally be appropriate if you're are comparing with a method that has a very fast encode, and in Figure 19, SwinT models are strictly slower at encoding than Conv models.

In Table 3 the SwinT entropy transform models h_s are slower than their Conv counterparts, is there an explanation for that? Is there performance to gain by using the Conv variants there and using SwinT exclusively for the image synthesis layers g_s?

Suggestions / Comments:
4.1 Training = "3.5M and 3.1M batches" => "3.5M and 3.1M steps" is a clearer description for describing how many training steps the network sees.

Footnote 11 in the Appendix is repeated three paragraphs above, feel free to remove one of them:
"We did not use the crop size 384 × 384 during the second stage as in the original paper because the
resolution of Vimeo dataset is 448×256. We found in our case increasing crop size from 256×256
to 384 × 256 in the second stage does not improve RD."

Do you have any timing for the SwinT vs. Conv models without a GPU? Do SwinT models have a runtime advantage with using a larger working set (max GPU memory) of very fast GPU memory vs the Conv models that don't have as high of a max memory usage?



**Summary Of The Paper:**

In this paper, the authors show that applying the Swin Transformer to both neural and image video compression show significant rate distortion gains over Conv Transformer methods and provide an analysis of latents and ablasion studies to understand the difference in the two methods.

**Summary Of The Review:**

This paper shows that Transformer based transforms can replace Conv based transforms in image and video compression, and simultaneously achieve better RD performance at much faster decode times. The results are strong across datasets for image and video compression and multiple entropy modeling techniques. The abundance of evidence and details, along with the state of the art results in performance make this a very strong paper.

---

> ### Author Response · Authors · 2021-11-17
> **Response to reviewer (2/2)**
>
> > Suggestions / Comments:
> 4.1 Training = "3.5M and 3.1M batches" => "3.5M and 3.1M steps" is a clearer description for describing how many training steps the network sees.
>
>
> >Footnote 11 in the Appendix is repeated three paragraphs above, feel free to remove one of them: "We did not use the crop size 384 × 384 during the second stage as in the original paper because the resolution of Vimeo dataset is 448×256. We found in our case increasing crop size from 256×256 to 384 × 256 in the second stage does not improve RD."
>
> Fixed, thanks.
>
> > Do you have any timing for the SwinT vs. Conv models without a GPU? Do SwinT models have a runtime advantage with using a larger working set (max GPU memory) of very fast GPU memory vs the Conv models that don't have as high of a max memory usage?
>
> As argued in the response to `Reviwer EyC9`, we believe neural codec should run on accelerated hardware. CPU timing might not provide much guidance in terms of future design iterations. Thus we choose not to profile neural codecs on CPU.
>
> The peak GPU memory of inference on single Kodak image takes less than 2GB, and less than 6.5 GB for resolution 1792×1024. There is 12GB memory on RTX 2080 used for profiling, thus GPU memory is not much a concern here.
>
> Although we are not experts in compiler or cuda, we think Convolution has been heavily optimized over the years with efficient kernels, while the accerlation for transformer/attention has just started recently. There is potentially more room for accelerating transformers in different hardware (GPU/TPU/NPU etc) in the coming years.

---

> ### Author Response · Authors · 2021-11-17
> **Response to reviewer (1/2)**
>
> Thanks for the detailed review and questions. Revised part is shown in `blue` in pdf.
>
> > In Figure 19, SwinT models are shown to be much slower at encoding than Conv models while much faster at decoding. Is there a possible shift of compute going on in the SwinT networks that aren't being taken advantage of in the Conv models?
>
> The encoder and decoder are symmetric in both transformer and Conv models. e.g. the inference time of SwinT encoder and SwinT decoder is close as shown in Fig 19 (a). The main reason for the different runtime of Conv encoder and Conv decoder is that Transposed Conv (in decoder) takes much more time than Conv (in encoder). We profiled for Conv-Hyperprior model coding Kodak images on RTX 2080 GPU. It shows that Transposed Conv (stride 2) with input shape (1, 320, 128, 192) and kernel size (320, 320, 5, 5) takes about 75969 $\mu s$, while Conv (stride 2) with input shape (1, 320, 256, 384) and kernel size (320, 320, 5, 5) takes about 10462 $\mu s$.
>
> It would be interesting to see if there is a competitive combination of asymmetric transformer-based encoder/decoder, or even heterogenous encoder/decoder. (although our experiment with different combinations of e.g. SwinT encoder & Conv decoder, or Conv encoder & SwinT decoder did not show promissing results)
>
> > The gap at low bitrates is amazing, the SwinT models heavily out perform the Conv models, but that is gain disappears at higher quality / rates, any intuition on why that is?
>
> Good question! We also observe the same trend for VTM-vs-BPG as SwinT-vs-Conv. This could be related to the fact that lossy compression at higher bitrate is approaching the setting of lossless compression, where there is just less room of coding gain with better predictions since the codec needs to encode the noise (not predictable) inherent in the data. This somehow agrees with the small performance improvement in generative modeling, e.g. relatively small improvement in negative log-likelihood, Fig 1 (a) of [8], or the small bit-per-dim gap in recent learned lossless image compression, Table 1 of [9].
>
> It is reassuring to see the bitrate saving of SwinT-ChARM w.r.t. VTM stays almost constant across the PSNRs (Fig 30 in our paper), which indicates our model achieves consistent coding efficiency as the heavily optimized conventional codec.
>
>  8. Kingma, Diederik P., et al. "Variational diffusion models." arXiv preprint arXiv:2107.00630 (2021).
>  9. van den Berg, Rianne, et al. "IDF++: Analyzing and improving integer discrete flows for lossless compression." International Conference on Learning Representations. 2020.
>
>
>
> > The image and video codec baselines were used in medium and with low/zero latency optimizations enabled. This will generally put classic codecs at a lower R/D in comparisons. This would generally be appropriate if you're are comparing with a method that has a very fast encode, and in Figure 19, SwinT models are strictly slower at encoding than Conv models.
>
> We agree that encoding complexity should be matched when comparing codecs. For image compression, encoding time of Swint/Conv models is much less than VTM (please check the table in reply to `Review EyC9`).
> Thus we believe the comparison is actually more favorable for traditional codecs in terms of encoding complexity, as pointed out by `Review EyC9` (fast encoding as a strength of neural codecs).
> For video compression, our model is a P-frame compression solution and is tested without rate-control, which, in ffmpeg, corresponds to `-tune zerolatency` setting.
> > In Table 3 the SwinT entropy transform models h_s are slower than their Conv counterparts, is there an explanation for that? Is there performance to gain by using the Conv variants there and using SwinT exclusively for the image synthesis layers g_s?
>
> Thanks for the detailed observation. The different relative runtime of $h_s$ and $g_s$ could be related to multiple reasons. According to our profiling, the runtime of Transponsed Conv becomes less expensive than Conv at lower feature resolution, e.g. 1025 $\mu s$ for Transposed Conv (stride 2) with input shape (1, 192, 8, 12) and kernel size (192, 192, 5, 5), and 3007 $\mu s$ for Conv (stride 2) with input shape (1, 192, 16, 24) and kernel size (192, 192, 5, 5). The other reason could be the relative complexity of SwinT $h_s$ and Conv $h_s$ may be not aligned with that of SwinT $g_s$ and Conv $g_s$. Note that the runtime for hyper decoder $h_s$ is negligible compared to that of decoder $g_s$ and entropy coding of the latent $\hat{y}$.
>
> The first model we tried is actually only replacing enc/dec pair $(g_a, g_s)$ with SwinT. We later further replaced hyper enc/dec pair $(h_a, h_s)$ with SwinT, which leads to a small gain in RD, but not much. Note that in our SwinT-SSF, we only replaced $(g_a, g_s)$ with SwinT, still using Conv for $(h_a, h_s)$.

---

### Official Review · Reviewer_aEdf · 2021-11-02

**Correctness:** 4
**Technical Novelty And Significance:** 3
**Empirical Novelty And Significance:** 3
**Recommendation:** 6
**Confidence:** 4

**Main Review:**

The proposed transformer-based framework shows good performance, proving that the transformer is a good replacement for typical convolution for compression tasks.

The experiments are rich, and enough evidences from many different aspects show that the transformer is just better than custom convolution.

However, a more important question is why transformer is better for image compression network, there is a lack of good organization and dig-in analysis over these many experiment results.


**Summary Of The Paper:**

This paper proposed to replace the typical cnn-based transform in image and video compression networks by Swin-transform. Experiments show its better performance and lower computational complexity over cnn-basd methods. Some extensive analysis are conducted to explore the differences between convolution and transformers.

**Summary Of The Review:**

The contribution of this paper is simple and direct, and the results also show good performance. But since transformer itself is not new, I wish to see deeper analyze over the experiments. So my recommendation is marginally below the borderline.

---

> ### Author Response · Authors · 2021-11-17
> **Response to reviewer**
>
> Thank you for the review.
> We totally agree that it is of great technical interest to form a deep understanding of the fundamental advantage of transformer-based architecture compared with convolutional ones. This in itself is a difficult problem and requires continuous and in-depth study, such as the concurrent attempts [4-6].
>
> In this work, we try to examine and analyze the difference of the two in data compression setting. Our analyses reveal strong evidence that the gain of R-D can be partially explained by transformer’s ability to better decorrelate spatial content, its flexibility in adapting the effective receptive field, and more localized decoder response to a single spatial latent pixel. We also did various experiments in the ablation study to gauge the effectiveness of different components of the overall transformer architecture. It is our hope that these analyses, albeit limited, can inspire and trigger more follow up studies that can help better explain the advantage of transformer-based models and design better models.
>
> For revision (`orange` text in the appendix D.3-D5 of the reivsion pdf), we further add some analyses of latent channels for both Conv and SwinT models, including interesting phenomenon common to both models, and an unique feature to SwinT models.
>
>  - Rate distribution of latent channels: we observe an interesting cutoff behavior in the rate-vs-channel curve
>  - Centered Kernel Analysis (CKA) [7] for feature similarity between 320 SwinT latent channels and 320 Conv latent channels
>  - Identification of SwinT and Conv latent chanenls with CKA
>  - Visualization of channel-wise progressive decoding, identified 3 important latent channels exist in both type of models, which are responsible for luma and two color components of the reconstruction, and one unique channel in SwinT latents which is of coarser scale than other latent channels, and important for the reconstruction with extremely few bits.
>
>
> Regarding the comment on the lack of good organization, we would appreciate and welcome any specific concerns/criticisms and would be happy to learn how to improve the presentation or clarity of the current work.
>
>  4. Tuli, Shikhar, et al. "Are Convolutional Neural Networks or Transformers more like human vision?." arXiv preprint arXiv:2105.07197 (2021).
>  5.  Raghu, Maithra, et al. "Do Vision Transformers See Like Convolutional Neural Networks?." arXiv preprint arXiv:2108.08810 (2021).
>  6. Anomymous. "How Do Vision Transformers Work?" The Tenth International Conference on Learning Representations (under review) (2022)
>  7. Kornblith, Simon, et al. "Similarity of neural network representations revisited." International Conference on Machine Learning. PMLR, 2019.

---

> > ### Comment · Reviewer_aEdf · 2021-11-18
> > **see more analyses in the revision**
> >
> > I do agree that it is not easy to fully understand the advantages of transformer through experiments and I have seen more analyses in the revision. My recommendation now is above the acceptance threshold.

---

### Official Review · Reviewer_7pae · 2021-11-02

**Correctness:** 4
**Technical Novelty And Significance:** 3
**Empirical Novelty And Significance:** 4
**Recommendation:** 6
**Confidence:** 5

**Main Review:**

Strengths:
1. The paper is well organized and the authors provide very comprehensive results and analysis for SwinT in compression. For example, the analysis of ERF is quite interesting and provides some insights for the following works.
2. The experimental results are convincing. Although some latest works like Guo et al. achieve slightly better compression performance, the proposed approach is much neater.
3. Basically, this is the first transformed-based coding solution, although performance is somewhat expected considering the success in vision transformer. The overall architecture and implementation are meaningful.

Weaknesses:
Some related works or experiments are suggested.
(1) It would be better to provide results if SwinT based compression also uses more expensive entropy coding approaches. Could we further boost the compression performance?
(2) There are also some approaches for low-complexity entropy coding approaches like He et al., Checkerboard Context Model for Efficient Learned Image Compression.  If we can use this approach to replace the expensive entropy coding method, could we achieve a better performance-complexity trade-off even we use Conv based model?  Please provide some discussion.
(3) There are some related works in video compression, it is suggested for discussion or comparison.
(a) FVC: A New Framework Towards Deep Video Compression in Feature Space. Hu et al.
(b) ELF-VC: Efficient Learned Flexible-Rate Video Coding. Rippel et al.

**Summary Of The Paper:**

This paper introduces the swin-transformer for the learned image and video coding. Instead of developing more advanced and expensive entropy coding approaches, the authors focus on the transform network(i.e., networks of encoder/decoder) and improve the compression performance significantly.  The experimental results are impressive. Although introducing SwinT is not very new, the authors provide a neat and practical solution, which should be encouraged. The authors also have a good balance between the coding gain and complexity.

**Summary Of The Review:**

I think the paper could be accepted considering its effective design for transformed-based compression as well as the comprehensive analysis.

---

> ### Author Response · Authors · 2021-11-17
> **Response to Reviewer**
>
> Thanks for the review. Revised part is showed in `magenta` in the paper pdf.
> > (1) It would be better to provide results if SwinT based compression also uses more expensive entropy coding approaches. Could we further boost the compression performance?
>
> We have also thought about this idea, but it does not align with the focus of this work, which is to achieve a better balance in terms of rate-distortion-complexity trade-off, instead of best RD without worrying about decoding complexity. Thus we decided not to further look into this direction.
>
> > (2) There are also some approaches for low-complexity entropy coding approaches like He et al., Checkerboard Context Model for Efficient Learned Image Compression. If we can use this approach to replace the expensive entropy coding method, could we achieve a better performance-complexity trade-off even we use Conv based model? Please provide some discussion.
>
> Thanks for pointing out this reference [1] to us. Our analyses on latent spatial correlation and effective receptive field agree with the observation in this paper that checkboard context is probably one of the most important contexts for image compression.
> Checkboard context can be interpretated as 2 groups of spatial latents (defined by the checkboard pattern) decoded sequentially, instead of 10 groups of (channel-wise) latents as used in ChARM [2]. We will add a short discussion in Section 4.3.
>
> Not sure if we understand your question of using checkerboard context prior and conv-based model? do you mean Conv-based transforms? If so, it would be the same design as in the mentioned paper, which uses the transforms from [3]. The results in Figure 1 of [1]
> shows similar performance compared to [2]. The comparison of [2] and the SwinT-based models can be found in Fig 30 of our paper.
> It would be interesting to find out how the combination of checkboard context prior and SwinT-based transfoms would work.
>
>  1. Dailan He, Yaoyan Zheng, Baocheng Sun, Yan Wang, Hongwei Qin. Checkerboard Context Model for Efficient Learned Image Compression. CVPR 2021
>  2. Minnen, David, and Saurabh Singh. "Channel-wise autoregressive entropy models for learned image compression." 2020 IEEE International Conference on Image Processing (ICIP). IEEE, 2020.
>  3. Cheng, Zhengxue, et al. "Learned image compression with discretized gaussian mixture likelihoods and attention modules." Proceedings of the IEEE/CVF Conference on Computer Vision and Pattern Recognition. 2020.
>
> > (3) There are some related works in video compression, it is suggested for discussion or comparison.
>
> Indeed, there are a plethora of work proposing various types of ConvNet-based architectures for neural video compression. The Scale-space-flow model is chosen as a case-study for neural video compression to show that the gain obtained in neural image compression setting can be readily transferred to video setting, and we hope it can trigger more follow up studies that continue to drive advancements in rate-distortion-complexity tradeoff. We will add more references regarding existing neural video compression works and discuss their relevance. Thanks for the suggestion.

---

### Official Review · Reviewer_EyC9 · 2021-11-02

**Correctness:** 3
**Technical Novelty And Significance:** 2
**Empirical Novelty And Significance:** 4
**Recommendation:** 8
**Confidence:** 4

**Details Of Ethics Concerns:**

No ethical concerns.

**Main Review:**

The primary strength of the paper is reaching the empirical milestone of outperforming VTM in terms of both rate-distortion and decode runtime. This is achieved by merging existing models: swin transformers (SwinT) for transforms and a channel-wise autoregressive model (ChARM) for the entropy model.

Another strength is the thorough evaluation in terms of transform variants (SwinT vs. convolution-based), entropy models (ChARM vs. a hyperprior), architecture size (number of layers and channel depths), and application to p-frame encoding for video compression. The authors also include a thorough range of ablation experiments, look at the ability for different transforms to generate a spatially independent latent representation, and explore the effective receptive field.

One weakness of the paper is in the runtime comparison between VTM and the various learned methods, though this is generally a difficult comparison to make. The difficulty arises because VTM runs on CPU (the authors state that an i9-9940X was used), while neural networks are typically run on a GPU or TPU (the authors say that an RTX 2080 Ti was used). This leads to an apples-to-oranges comparison that should be stated more explicitly. Decode times on the CPU would also strengthen the paper.

In addition, the runtime of VTM varies with encoding quality. At least based on our tests (caveat: on a different CPU and with a different version of VTM), VTM can decode a 768x512 image in anywhere from 20ms (at the lowest quality level) up to 230ms for the highest quality (note that these numbers include disk I/O). Based on this, a more accurate claim is that SwinT-Charm decodes faster than VTM at some quality levels (very high ones), but can be much slower for high compression rates.

Although the authors don't mention it, VTM can actually be very slow to *encode* an image at a very high quality level (on the order of dozens of seconds). I agree that decode speed is most important for typical applications (since images are often encoded once and decoded millions of times), the relatively fast encode speed of the proposed model can be seen as a strength.

**Summary Of The Paper:**

This paper addresses the problem of improving rate-distortion (RD) performance for learned image and video compression without ignoring runtime. Much of the literature on learned compression improves RD performance with more entropy models. Especially for autoregressive models, these models can have excessively slow decode times. This paper uses a less expensive (and generally less powerful) entropy model and explores the use of a transformer in the encoder and decoder transforms instead.

The result is a much faster model that still achieves near-SOTA rate-distortion performance. In particular, the SwinT-ChARM model described in this paper outperforms VVC (via VTM 12.1 a recent version of the reference implementation for H.266) with faster decode times. As far as I know, this is a milestone for learned image compression (other methods have outperformed in terms of RD but only with much slower decoders).


**Summary Of The Review:**

Despite some reservations about the technical novelty and the runtime comparison, I still think this paper deserves an "accept, good paper" recommendation.

This conclusion is based on the strong empirical results and the thorough evaluation. The results are also somewhat surprising to me, which is an indicator of important research. I would have expected transformers to improve RD performance compared to conv-nets, but I expected a smaller difference and for runtime to *increase*, not *decrease*.

I also think that if I were to start working on learned image/video compression, I would use the model in this paper as a baseline.

---

> ### Author Response · Authors · 2021-11-17
> **Response to Reviewer**
>
> Thanks for sharing the excitement about the results as we do. Revised part is showed in `teal` in the paper pdf.
> > One weakness of the paper is in the runtime comparison between VTM and the various learned methods, though this is generally a difficult comparison to make. The difficulty arises because VTM runs on CPU (the authors state that an i9-9940X was used), while neural networks are typically run on a GPU or TPU (the authors say that an RTX 2080 Ti was used). This leads to an apples-to-oranges comparison that should be stated more explicitly. Decode times on the CPU would also strengthen the paper.
>
> We appreciate the point that fair comparison of decoding time is difficulty, especially considering that classic codecs and neural codecs are designed targeting different types of hardware. This point is worth highlighting and we will state it more explicitly in the paper.
> It is still early to know what types of hardware neural codecs will be deployed on eventually, but we think it should run on accelerators, such as GPU/TPU/NPU, FPGA, etc. Thus we did not report CPU runtime in the paper, as CPU timing might not provide valuable guidance for future model design.
>
> > In addition, the runtime of VTM varies with encoding quality. At least based on our tests (caveat: on a different CPU and with a different version of VTM), VTM can decode a 768x512 image in anywhere from 20ms (at the lowest quality level) up to 230ms for the highest quality (note that these numbers include disk I/O). Based on this, a more accurate claim is that SwinT-Charm decodes faster than VTM at some quality levels (very high ones), but can be much slower for high compression rates.
>
> Indeed, the decoding time is a function of quality levels. To echo the reviewer's point, in the table below we run Kodak through VTM12.1 with QP values of 16 and 40 as two extreme quality settings, and can observe that the average decoding time ranges from 0.155s to 0.295s (i9-9940x). The VTM running time reported in Figure~1 is taken from QP=28, with a bpp of 0.7844 that is close to what we get from the beta=0.001 neural-network model that we compare with.
>
> | QP | BPP | PSNR(dB) | Decoding time (s) | Encoding time (s) |
> |----|-----|----------|-------------------|-------------------|
> |16	| 2.5441 |	44.09 |	0.284 |	300.47 |
> |28	| 0.7844 |	36.76 |	0.249 |	118.85 |
> |40	| 0.1557 |	29.51 |	0.149 |	71.10 |
>
> It is because of this difficulty in coming up a fair comparison that we decide not to make any strong claim as to which one is faster, but rather just state that the proposed solution runs at a comparable speed as VTM.
>
>
> > Although the authors don't mention it, VTM can actually be very slow to encode an image at a very high quality level (on the order of dozens of seconds). I agree that decode speed is most important for typical applications (since images are often encoded once and decoded millions of times), the relatively fast encode speed of the proposed model can be seen as a strength.
>
> We thank the reviewer for pointing out the fact that VTM runs slow in encoding process (evident from the table above) and the relatively fast encode speed of the proposed model can be seen as a strength. We will make sure to mention it in the paper. We also mentioned in the end of the first subsection of Section 4.2 that additional gains can be achieved by exploiting this tolerance of encoding time (e.g. test time latent optimization or instance adaptation).

---

### Official Review · Reviewer_xjay · 2021-11-06

**Correctness:** 4
**Technical Novelty And Significance:** 4
**Empirical Novelty And Significance:** 3
**Recommendation:** 8
**Confidence:** 3

**Main Review:**

1. Nice exploration in extending Swin-Transformer to a decoder setting and build Swin-transformer based neural image codecs. The experiments show that Swin achieved better rate-distortion performance with lower complexity than existing solutions.

2. The authors further deomonstrated the effectiveness of Swin-transformer in video compression by enhancing scale-space-flow, a popular neural P-frame codec.

3. For the detailed experiments, they show that the inference time of SwinT decoder is less than that of Conv decoder, also Swin has effective receptive field, incur less redundancy across different spatial latent locations, and progressive decoding

**Summary Of The Paper:**

The authors extended Swin-Transformer to neural image codecs and demonstrated that Swin-transformers - SwinT-ChARM (proposed) can achieve better compression efficiency than convolutional neural networks (ConvNets), while requiring fewer parameters and shorter decoding time.

**Summary Of The Review:**

The authors did a novel exploration in using Swin-Transformer for transform coding and proved that SwinT-ChARM (the authors proposed) demonstrated better compression efficiency as opposed to CNN counterparts. It is a nice exploration and provides new perspective in image codecs.

---

> ### Author Response · Authors · 2021-11-17
> **Response to reviewer**
>
> Thank you for your review. We appreciate your positive feedback.

---

### Public Comment · ~Yinhao_Zhu1 · 2022-07-18
**RD numbers of our SwinT-based models on standard testsets (from authors)**

|                  | Kodak       |             |  | CLIC test 2021 |             |  | JPEG AI test |             |  | Tecnick test |             |
|------------------|-------------|-------------|--|----------------|-------------|--|--------------|-------------|--|--------------|-------------|
|                  | bpp         | psnr_rgb    |  | bpp            | psnr_rgb    |  | bpp          | psnr_rgb    |  | bpp          | psnr_rgb    |
| swint-hyperprior | 2.533632914 | 44.15740458 |  | 1.825757332    | 44.94729258 |  | 2.46894912   | 42.12494898 |  | 2.066397476  | 43.46328442 |
|                  | 1.561948988 | 40.84309435 |  | 1.096847388    | 41.92048747 |  | 1.577080075  | 39.58504963 |  | 1.19615509   | 40.69666943 |
|                  | 0.957267761 | 37.81027428 |  | 0.641195697    | 39.15450211 |  | 0.961770789  | 36.9283396  |  | 0.694435501  | 38.20763828 |
|                  | 0.51915741  | 34.49336886 |  | 0.331475584    | 36.20349503 |  | 0.514976263  | 33.88311493 |  | 0.374988327  | 35.53226341 |
|                  | 0.275058322 | 31.5492026  |  | 0.182496239    | 33.61250734 |  | 0.284378183  | 31.15561926 |  | 0.216649323  | 33.04803041 |
|                  |             |             |  |                |             |  |              |             |  |              |             |
| swint-charm      | 2.389131334 | 44.00882594 |  | 1.72489615     | 44.87883212 |  | 2.320349659  | 41.93447828 |  | 1.938998718  | 43.32239704 |
|                  | 1.504291111 | 40.92698352 |  | 1.058873131    | 42.02154992 |  | 1.527481644  | 39.71072555 |  | 1.150656281  | 40.809338   |
|                  | 0.923138089 | 37.87930886 |  | 0.618541513    | 39.21881892 |  | 0.928183047  | 37.00336361 |  | 0.667811432  | 38.29026237 |
|                  | 0.502814399 | 34.57582458 |  | 0.319520753    | 36.28533767 |  | 0.498641956  | 33.96892929 |  | 0.36115654   | 35.62436407 |
|                  | 0.264351739 | 31.60599279 |  | 0.172998007    | 33.69929988 |  | 0.274217245  | 31.24082124 |  | 0.206610565  | 33.14379164 |

Note that if the image size can not be divided by 256, center crop to largest possible multiple of 256 is applied to the image before evaluating models (including the traditional codecs).

---

### Decision · Program_Chairs · 2022-01-20

**Decision:**

Accept (Poster)

**Comment:**

The paper aims to improve image and video compression keeping in mind computation cost. In this regard, authors propose variants of Swin-Transformer for image and video coding. The experimental results shows that Transformer based transforms can replace Conv based transforms in image and video compression, and simultaneously achieving better rate-distortion performance at much faster decode times, i.e. resulting in a better rate-distortion-computation trade-off. We thank the reviewers and authors for engaging in an active discussion. The reviewers found some results to be surprising (in a good way) and are in a consensus that the empirical results are strong across datasets for image and video compression. For completeness, the authors should provide FLOPs or CPU runtimes in the final version so that one can compare to methods like VTM even if CPU is not the desired hardware for proposed method.